# Rényi Divergence Variational Inference

**Yingzhen Li**
University of Cambridge
Cambridge, CB2 1PZ, UK
yl494@cam.ac.uk

**Richard E. Turner**
University of Cambridge
Cambridge, CB2 1PZ, UK
ret26@cam.ac.uk

## Abstract

This paper introduces the *variational Rényi bound (VR)* that extends traditional variational inference to Rényi's $\alpha$-divergences. This new family of variational methods unifies a number of existing approaches, and enables a smooth interpolation from the evidence lower-bound to the log (marginal) likelihood that is controlled by the value of $\alpha$ that parametrises the divergence. The reparameterization trick, Monte Carlo approximation and stochastic optimisation methods are deployed to obtain a tractable and unified framework for optimisation. We further consider negative $\alpha$ values and propose a novel variational inference method as a new special case in the proposed framework. Experiments on Bayesian neural networks and variational auto-encoders demonstrate the wide applicability of the VR bound.

## 1 Introduction

Approximate inference, that is approximating posterior distributions and likelihood functions, is at the core of modern probabilistic machine learning. This paper focuses on optimisation-based approximate inference algorithms, popular examples of which include variational inference (VI), variational Bayes (VB) [1, 2] and expectation propagation (EP) [3, 4]. Historically, VI has received more attention compared to other approaches, although EP can be interpreted as iteratively minimising a set of local divergences [5]. This is mainly because VI has elegant and useful theoretical properties such as the fact that it proposes a lower-bound of the log-model evidence. Such a lower-bound can serve as a surrogate to both maximum likelihood estimation (MLE) of the hyper-parameters and posterior approximation by Kullback-Leibler (KL) divergence minimisation.

Recent advances of approximate inference follow three major trends. First, scalable methods, e.g. stochastic variational inference (SVI) [6] and stochastic expectation propagation (SEP) [7, 8], have been developed for datasets comprising millions of datapoints. Recent approaches [9, 10, 11] have also applied variational methods to coordinate parallel updates arising from computations performed on chunks of data. Second, Monte Carlo methods and black-box inference techniques have been deployed to assist variational methods, e.g. see [12, 13, 14, 15] for VI and [16] for EP. They all proposed ascending the Monte Carlo approximated variational bounds to the log-likelihood using noisy gradients computed with automatic differentiation tools. Third, tighter variational lower-bounds have been proposed for (approximate) MLE. The importance weighted auto-encoder (IWAE) [17] improved upon the variational auto-encoder (VAE) [18, 19] framework, by providing tighter lower-bound approximations to the log-likelihood using importance sampling. These recent developments are rather separated and little work has been done to understand their connections.

In this paper we try to provide a unified framework from an energy function perspective that encompasses a number of recent advances in variational methods, and we hope our effort could potentially motivate new algorithms in the future. This is done by extending traditional VI to Rényi's $\alpha$-divergence [20], a rich family that includes many well-known divergences as special cases. After reviewing useful properties of Rényi divergences and the VI framework, we make the following contributions:

Table 1: Special cases in the Rényi divergence family.

| $\alpha$ | Definition | Notes |
|---|---|---|
| $\alpha \to 1$ | $\int p(\boldsymbol{\theta}) \log \frac{p(\boldsymbol{\theta})}{q(\boldsymbol{\theta})} d\boldsymbol{\theta}$ | *Kullback-Leibler (KL) divergence*, used in VI (KL$[q\|p]$) and EP (KL$[p\|q]$) |
| $\alpha = 0.5$ | $-2\log(1 - \text{Hel}^2[p\|q])$ | function of the square *Hellinger distance* |
| $\alpha \to 0$ | $-\log \int_{p(\boldsymbol{\theta})>0} q(\boldsymbol{\theta}) d\boldsymbol{\theta}$ | zero when $\text{supp}(q) \subseteq \text{supp}(p)$ (not a divergence) |
| $\alpha = 2$ | $-\log(1 - \chi^2[p\|q])$ | proportional to the $\chi^2$-divergence |
| $\alpha \to +\infty$ | $\log \max_{\boldsymbol{\theta} \in \Theta} \frac{p(\boldsymbol{\theta})}{q(\boldsymbol{\theta})}$ | *worst-case regret* in *minimum description length principle* [24] |

- We introduce the *variational Rényi bound* (VR) as an extension of VI/VB. We then discuss connections to existing approaches, including VI/VB, VAE, IWAE [17], SEP [7] and black-box alpha (BB-$\alpha$) [16], thereby showing the richness of this new family of variational methods.
- We develop an optimisation framework for the VR bound. An analysis of the bias introduced by stochastic approximation is also provided with theoretical guarantees and empirical results.
- We propose a novel approximate inference algorithm called *VR-max* as a new special case. Evaluations on VAEs and Bayesian neural networks show that this new method is often comparable to, or even better than, a number of the state-of-the-art variational methods.

## 2 Background

This section reviews Rényi's $\alpha$-divergence and variational inference upon which the new framework is based. Note that there exist other $\alpha$-divergence definitions [21, 22] (see appendix). However we mainly focus on Rényi's definition as it enables us to derive a new class of variational lower-bounds.

### 2.1 Rényi's $\alpha$-divergence

We first review Rényi's $\alpha$-divergence [20, 23]. Rényi's $\alpha$-divergence, defined on $\{\alpha : \alpha > 0, \alpha \neq 1, |D_\alpha| < +\infty\}$, measures the "closeness" of two distributions $p$ and $q$ on a random variable $\boldsymbol{\theta} \in \Theta$:

$$D_\alpha[p\|q] = \frac{1}{\alpha - 1} \log \int p(\boldsymbol{\theta})^\alpha q(\boldsymbol{\theta})^{1-\alpha} d\boldsymbol{\theta}. \tag{1}$$

The definition is extended to $\alpha = 0, 1, +\infty$ by continuity. We note that when $\alpha \to 1$ the Kullback-Leibler (KL) divergence is recovered, which plays a crucial role in machine learning and information theory. Some other special cases are presented in Table 1. The method proposed in this work also considers $\alpha \leq 0$ (although (1) is no longer a divergence for these $\alpha$ values), and we include from [23] some useful properties for forthcoming derivations.

**Proposition 1.** *(Monotonicity) Rényi's $\alpha$-divergence definition (1), extended to negative $\alpha$, is **continuous** and **non-decreasing** on $\alpha \in \{\alpha : -\infty < D_\alpha < +\infty\}$.*

**Proposition 2.** *(Skew symmetry) For $\alpha \notin \{0, 1\}$, $D_\alpha[p\|q] = \frac{\alpha}{1-\alpha} D_{1-\alpha}[q\|p]$. This implies $D_\alpha[p\|q] \leq 0$ for $\alpha < 0$. For the limiting case $D_{-\infty}[p\|q] = -D_{+\infty}[q\|p]$.*

A critical question that is still in active research is how to choose a divergence in this rich family to obtain optimal solution for a particular application, an issue which is discussed in the appendix.

### 2.2 Variational inference

Next we review the variational inference algorithm [1, 2] using posterior approximation as a running example. Consider observing a dataset of $N$ i.i.d. samples $\mathcal{D} = \{\boldsymbol{x}_n\}_{n=1}^N$ from a probabilistic model $p(\boldsymbol{x}|\boldsymbol{\theta})$ parametrised by a random variable $\boldsymbol{\theta}$ that is drawn from a prior $p_0(\boldsymbol{\theta})$. Bayesian inference involves computing the posterior distribution of the parameters given the data,

$$p(\boldsymbol{\theta}|\mathcal{D}, \boldsymbol{\varphi}) = \frac{p(\boldsymbol{\theta}, \mathcal{D}|\boldsymbol{\varphi})}{p(\mathcal{D}|\boldsymbol{\varphi})} = \frac{p_0(\boldsymbol{\theta}|\boldsymbol{\varphi}) \prod_{n=1}^N p(\boldsymbol{x}_n|\boldsymbol{\theta}, \boldsymbol{\varphi})}{p(\mathcal{D}|\boldsymbol{\varphi})}, \tag{2}$$

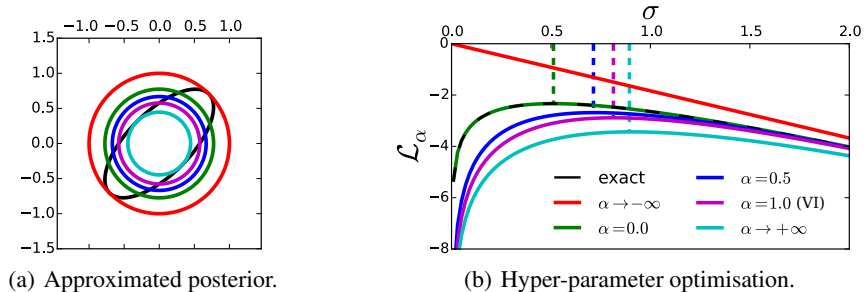

(a) Approximated posterior.  (b) Hyper-parameter optimisation.

Figure 1: Mean-Field approximation for Bayesian linear regression. In this case $\boldsymbol{\varphi} = \sigma$ the observation noise variance. The bound is tight as $\sigma \to +\infty$, biasing the VI solution to large $\sigma$ values.

where $p(\mathcal{D}|\boldsymbol{\varphi}) = \int p_0(\boldsymbol{\theta}|\boldsymbol{\varphi}) \prod_{n=1}^{N} p(\boldsymbol{x}_n|\boldsymbol{\theta}, \boldsymbol{\varphi}) d\boldsymbol{\theta}$ is called marginal likelihood or model evidence. The hyper-parameters of the model are denoted as $\boldsymbol{\varphi}$ which might be omitted henceforth for notational ease. For many powerful models the exact posterior is typically intractable, and approximate inference introduces an approximation $q(\boldsymbol{\theta})$ in some tractable distribution family $\mathcal{Q}$ to the exact posterior. One way to obtain this approximation is to minimise the KL divergence $\text{KL}[q(\boldsymbol{\theta})||p(\boldsymbol{\theta}|\mathcal{D})]$, which is also intractable due the difficult term $p(\mathcal{D})$. Variational inference (VI) sidesteps this difficulty by considering an equivalent optimisation problem that maximises the *variational lower-bound*:

$$\mathcal{L}_{\text{VI}}(q; \mathcal{D}, \boldsymbol{\varphi}) = \log p(\mathcal{D}|\boldsymbol{\varphi}) - \text{KL}[q(\boldsymbol{\theta})||p(\boldsymbol{\theta}|\mathcal{D}, \boldsymbol{\varphi})] = \mathbb{E}_q \left[ \log \frac{p(\boldsymbol{\theta}, \mathcal{D}|\boldsymbol{\varphi})}{q(\boldsymbol{\theta})} \right]. \quad (3)$$

The variational lower-bound can also be used to optimise the hyper-parameters $\boldsymbol{\varphi}$.

To illustrate the approximation quality of VI we present a mean-field approximation example to Bayesian linear regression in Figure 1(a) (in magenta). Readers are referred to the appendix for details, but essentially a factorised Gaussian approximation is fitted to the true posterior, a correlated Gaussian in this case. The approximation recovers the posterior mean correctly, but is over-confident. Moreover, as $\mathcal{L}_{\text{VI}}$ is the difference between the marginal likelihood and the KL divergence, hyper-parameter optimisation can be biased away from the exact MLE towards the region of parameter space where the KL term is small [25] (see Figure 1(b)).

## 3 Variational Rényi bound

Recall from Section 2.1 that the family of Rényi divergences includes the KL divergence. Perhaps variational free-energy approaches can be generalised to the Rényi case? Consider approximating the exact posterior $p(\boldsymbol{\theta}|\mathcal{D})$ by minimizing Rényi's $\alpha$-divergence $\text{D}_\alpha[q(\boldsymbol{\theta})||p(\boldsymbol{\theta}|\mathcal{D})]$ for some selected $\alpha > 0$. Now we consider the equivalent optimization problem $\max_{q \in \mathcal{Q}} \log p(\mathcal{D}) - \text{D}_\alpha[q(\boldsymbol{\theta})||p(\boldsymbol{\theta}|\mathcal{D})]$, and when $\alpha \neq 1$, whose objective can be rewritten as

$$\mathcal{L}_\alpha(q; \mathcal{D}) := \frac{1}{1-\alpha} \log \mathbb{E}_q \left[ \left( \frac{p(\boldsymbol{\theta}, \mathcal{D})}{q(\boldsymbol{\theta})} \right)^{1-\alpha} \right]. \quad (4)$$

We name this new objective the *variational Rényi (VR) bound*. Importantly the above definition can be extend to $\alpha \leq 0$, and the following theorem is a direct result of Proposition 1.

**Theorem 1.** *The objective $\mathcal{L}_\alpha(q; \mathcal{D})$ is **continuous** and **non-increasing** on $\alpha \in \{\alpha : |\mathcal{L}_\alpha| < +\infty\}$. Especially for all $0 < \alpha_+ < 1$ and $\alpha_- < 0$,*

$$\mathcal{L}_{VI}(q; \mathcal{D}) = \lim_{\alpha \to 1} \mathcal{L}_\alpha(q; \mathcal{D}) \leq \mathcal{L}_{\alpha_+}(q; \mathcal{D}) \leq \mathcal{L}_0(q; \mathcal{D}) \leq \mathcal{L}_{\alpha_-}(q; \mathcal{D})$$

*Also $\mathcal{L}_0(q; \mathcal{D}) = \log p(\mathcal{D})$ if and only if the support $\text{supp}(p(\boldsymbol{\theta}|\mathcal{D})) \subseteq \text{supp}(q(\boldsymbol{\theta}))$.*

Theorem 1 indicates that the VR bound can be useful for model selection by sandwiching the marginal likelihood with bounds computed using positive and negative $\alpha$ values, which we leave to future work. In particular $\mathcal{L}_0 = \log p(\mathcal{D})$ under the mild assumption that $q$ is supported where the exact

posterior is supported. This assumption holds for many commonly used distributions, e.g. Gaussians are supported on the entire space, and in the following we assume that this condition is satisfied.

Choosing different alpha values allows the approximation to balance between zero-forcing ($\alpha \to +\infty$, when using uni-modal approximations it is usually called mode-seeking) and mass-covering ($\alpha \to -\infty$) behaviour. This is illustrated by the Bayesian linear regression example, again in Figure 1(a). First notice that $\alpha \to +\infty$ (in cyan) returns non-zero uncertainty estimates (although it is more over-confident than VI) which is different from the maximum a posteriori (MAP) method that only returns a point estimate. Second, setting $\alpha = 0.0$ (in green) returns $q(\boldsymbol{\theta}) = \prod_i p(\theta_i|\mathcal{D})$ and the exact marginal likelihood $\log p(\mathcal{D})$ (Figure 1(b)). Also the approximate MLE is less biased for $\alpha = 0.5$ (in blue) since now the tightness of the bound is less hyper-parameter dependent.

## 4   The VR bound optimisation framework

This section addresses several issues of the VR bound optimisation by proposing further approximations. First when $\alpha \neq 1$, the VR bound is usually just as intractable as the marginal likelihood for many useful models. However Monte Carlo (MC) approximation is applied here to extend the set of models that can be handled. The resulting method can be applied to any model that MC-VI [12, 13, 14, 15] is applied to. Second, Theorem 1 suggests that the VR bound is to be minimised when $\alpha < 0$, which performs disastrously in MLE context. As we shall see, this issue is solved also by the MC approximation under certain conditions. Third, a mini-batch training method is developed for large-scale datasets in the posterior approximation context. Hence the proposed optimisation framework of the VR bound enables tractable application to the same class of models as SVI.

### 4.1   Monte Carlo approximation of the VR bound

Consider learning a latent variable model with MLE as a running example, where the model is specified by a conditional distribution $p(\boldsymbol{x}|\boldsymbol{h}, \boldsymbol{\varphi})$ and a prior $p(\boldsymbol{h}|\boldsymbol{\varphi})$ on the latent variables $\boldsymbol{h}$. Examples include models treated by the variational auto-encoder (VAE) approach [18, 19] that parametrises the likelihood with a (deep) neural network. MLE requires $\log p(\boldsymbol{x})$ which is obtained by marginalising out $\boldsymbol{h}$ and is often intractable, so the VR bound is considered as an alternative optimisation objective. However instead of using exact bounds, a simple Monte Carlo (MC) method is deployed, which uses finite samples $\boldsymbol{h}_k \sim q(\boldsymbol{h}|\boldsymbol{x}), k = 1, ..., K$ to approximate $\mathcal{L}_\alpha \approx \hat{\mathcal{L}}_{\alpha,K}$:

$$\hat{\mathcal{L}}_{\alpha,K}(q; \boldsymbol{x}) = \frac{1}{1-\alpha} \log \frac{1}{K} \sum_{k=1}^{K} \left[ \left( \frac{p(\boldsymbol{h}_k, \boldsymbol{x})}{q(\boldsymbol{h}_k|\boldsymbol{x})} \right)^{1-\alpha} \right]. \tag{5}$$

The importance weighted auto-encoder (IWAE) [17] is a special case of this framework with $\alpha = 0$ and $K < +\infty$. But unlike traditional VI, here the MC approximation is biased. Fortunately we can characterise the bias by the following theorems proved in the appendix.

**Theorem 2.** *Assume* $\mathbb{E}_{\{\boldsymbol{h}_k\}_{k=1}^{K}}[\|\hat{\mathcal{L}}_{\alpha,K}(q; \boldsymbol{x})\|] < +\infty$ *and* $|\mathcal{L}_\alpha| < +\infty$. *Then* $\mathbb{E}_{\{\boldsymbol{h}_k\}_{k=1}^{K}}[\hat{\mathcal{L}}_{\alpha,K}(q; \boldsymbol{x})]$ *as a function of* $\alpha \in \mathbb{R}$ *and* $K \geq 1$ *is:*
*1)* ***non-decreasing*** *in* $K$ *for fixed* $\alpha \leq 1$, *and* ***non-increasing*** *in* $K$ *for fixed* $\alpha \geq 1$;
*2)* $\mathbb{E}_{\{\boldsymbol{h}_k\}_{k=1}^{K}}[\hat{\mathcal{L}}_{\alpha,K}(q; \boldsymbol{x})] \to \mathcal{L}_\alpha$ *as* $K \to +\infty$;
*3)* ***continuous*** *and* ***non-increasing*** *in* $\alpha$ *with fixed* $K$.

**Corollary 1.** *For finite* $K$, *either* $\mathbb{E}_{\{\boldsymbol{h}_k\}_{k=1}^{K}}[\hat{\mathcal{L}}_{\alpha,K}(q; \boldsymbol{x})] < \log p(\boldsymbol{x})$ *for all* $\alpha$, *or there exists* $\alpha_K \leq 0$ *such that* $\mathbb{E}_{\{\boldsymbol{h}_k\}_{k=1}^{K}}[\hat{\mathcal{L}}_{\alpha_K,K}(q; \boldsymbol{x})] = \log p(\boldsymbol{x})$ *and* $\mathbb{E}_{\{\boldsymbol{h}_k\}_{k=1}^{K}}[\hat{\mathcal{L}}_{\alpha,K}(q; \boldsymbol{x})] > \log p(\boldsymbol{x})$ *for all* $\alpha < \alpha_K$. *Also* $\alpha_K$ *is* ***non-decreasing*** *in* $K$ *if exists, with* $\lim_{K\to 1} \alpha_K = -\infty$ *and* $\lim_{K\to +\infty} \alpha_K = 0$.

The intuition behind the theorems is visualised in Figure 2(a). By definition, the exact VR bound is a lower-bound or upper-bound of $\log p(\boldsymbol{x})$ when $\alpha > 0$ or $\alpha < 0$, respectively. However the MC approximation $\mathbb{E}[\hat{\mathcal{L}}_{\alpha,K}]$ biases the estimate towards $\mathcal{L}_{\text{VI}}$, where the approximation quality can be improved using more samples. Thus for finite samples and under mild conditions, negative alpha values can potentially be used to improve the accuracy of the approximation, at the cost of losing the upper-bound guarantee. Figure 2(b) shows an empirical evaluation by computing the exact and the MC approximation of the Rényi divergences. In this example $p, q$ are 2-D Gaussian distributions with $\boldsymbol{\mu}_p = [0, 0]$, $\boldsymbol{\mu}_q = [1, 1]$ and $\boldsymbol{\Sigma}_p = \boldsymbol{\Sigma}_q = \boldsymbol{I}$. The sampling procedure is repeated

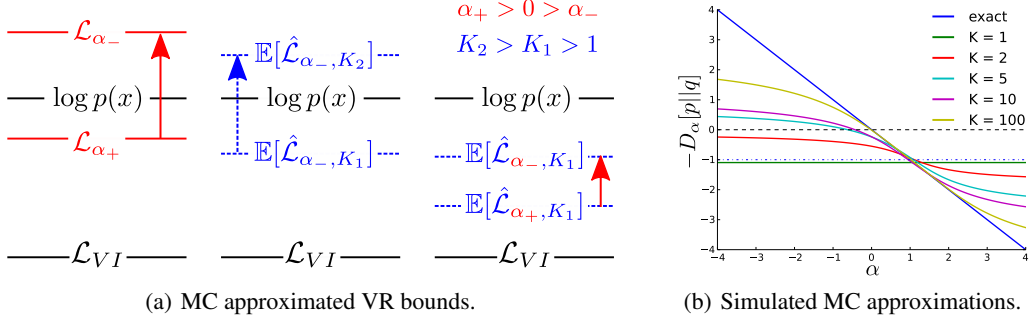

| (a) MC approximated VR bounds. | (b) Simulated MC approximations. |

Figure 2: (a) An illustration for the bounding properties of MC approximations to the VR bounds. (b) The bias of the MC approximation. Best viewed in colour and see the main text for details.

200 times to estimate the expectation. Clearly for $K = 1$ it is equivalent to an unbiased estimate of the KL-divergence for all $\alpha$ (even though now the estimation is biased for $D_\alpha$). For $K > 1$ and $\alpha < 1$, the MC method under-estimates the VR bound, and the bias decreases with increasing $K$. For $\alpha > 1$ the inequality is reversed also as predicted.

## 4.2    Unified implementation with the reparameterization trick

Readers may have noticed that $\mathcal{L}_{VI}$ has a different form compared to $\mathcal{L}_\alpha$ with $\alpha \neq 1$. In this section we show how to unify the implementation for all finite $\alpha$ settings using the *reparameterization trick* [13, 18] as an example. This trick assumes the existence of the mapping $\boldsymbol{\theta} = g_{\boldsymbol{\phi}}(\boldsymbol{\epsilon})$, where the distribution of the noise term $\boldsymbol{\epsilon}$ satisfies $q(\boldsymbol{\theta})d\boldsymbol{\theta} = p(\boldsymbol{\epsilon})d\boldsymbol{\epsilon}$. Then the expectation of a function $F(\boldsymbol{\theta})$ over distribution $q(\boldsymbol{\theta})$ can be computed as $\mathbb{E}_{q(\boldsymbol{\theta})}[F(\boldsymbol{\theta})] = \mathbb{E}_{p(\boldsymbol{\epsilon})}[F(g_{\boldsymbol{\phi}}(\boldsymbol{\epsilon}))]$. One prevalent example is the Gaussian reparameterization: $\boldsymbol{\theta} \sim \mathcal{N}(\boldsymbol{\mu}, \Sigma) \Rightarrow \boldsymbol{\theta} = \boldsymbol{\mu} + \Sigma^{\frac{1}{2}}\boldsymbol{\epsilon}, \boldsymbol{\epsilon} \sim \mathcal{N}(\mathbf{0}, I)$. Now we apply the reparameterization trick to the VR bound

$$\mathcal{L}_\alpha(q_{\boldsymbol{\phi}}; \boldsymbol{x}) = \frac{1}{1-\alpha}\log\mathbb{E}_{\boldsymbol{\epsilon}}\left[\left(\frac{p(g_{\boldsymbol{\phi}}(\boldsymbol{\epsilon}), \boldsymbol{x})}{q(g_{\boldsymbol{\phi}}(\boldsymbol{\epsilon}))}\right)^{1-\alpha}\right]. \tag{6}$$

Then the gradient of the VR bound w.r.t. $\boldsymbol{\phi}$ is (similar for $\boldsymbol{\varphi}$, see appendix for derivation)

$$\nabla_{\boldsymbol{\phi}}\mathcal{L}_\alpha(q_{\boldsymbol{\phi}}; \boldsymbol{x}) = \mathbb{E}_{\boldsymbol{\epsilon}}\left[w_\alpha(\boldsymbol{\epsilon}; \boldsymbol{\phi}, \boldsymbol{x})\nabla_{\boldsymbol{\phi}}\log\frac{p(g_{\boldsymbol{\phi}}(\boldsymbol{\epsilon}), \boldsymbol{x})}{q(g_{\boldsymbol{\phi}}(\boldsymbol{\epsilon}))}\right], \tag{7}$$

where $w_\alpha(\boldsymbol{\epsilon}; \boldsymbol{\phi}, \boldsymbol{x}) = \left(\frac{p(g_{\boldsymbol{\phi}}(\boldsymbol{\epsilon}), \boldsymbol{x})}{q(g_{\boldsymbol{\phi}}(\boldsymbol{\epsilon}))}\right)^{1-\alpha} \Big/ \mathbb{E}_{\boldsymbol{\epsilon}}\left[\left(\frac{p(g_{\boldsymbol{\phi}}(\boldsymbol{\epsilon}), \boldsymbol{x})}{q(g_{\boldsymbol{\phi}}(\boldsymbol{\epsilon}))}\right)^{1-\alpha}\right]$ denotes the normalised importance weight. One can show that this recovers the the stochastic gradients of $\mathcal{L}_{VI}$ by setting $\alpha = 1$ in (7) since now $w_1(\boldsymbol{\epsilon}; \boldsymbol{\phi}, \boldsymbol{x}) = 1$, which means the resulting algorithm unifies the computation for all finite $\alpha$ settings. For MC approximations, we use $K$ samples to approximately compute the weight $\hat{w}_{\alpha,k}(\boldsymbol{\epsilon}_k; \boldsymbol{\phi}, \boldsymbol{x}) \propto \left(\frac{p(g_{\boldsymbol{\phi}}(\boldsymbol{\epsilon}_k), \boldsymbol{x})}{q(g_{\boldsymbol{\phi}}(\boldsymbol{\epsilon}_k))}\right)^{1-\alpha}$, $k = 1, ..., K$, and the stochastic gradient becomes

$$\nabla_{\boldsymbol{\phi}}\hat{\mathcal{L}}_{\alpha,K}(q_{\boldsymbol{\phi}}; \boldsymbol{x}) = \sum_{k=1}^{K}\left[\hat{w}_{\alpha,k}(\boldsymbol{\epsilon}_k; \boldsymbol{\phi}, \boldsymbol{x})\nabla_{\boldsymbol{\phi}}\log\frac{p(g_{\boldsymbol{\phi}}(\boldsymbol{\epsilon}_k), \boldsymbol{x})}{q(g_{\boldsymbol{\phi}}(\boldsymbol{\epsilon}_k))}\right]. \tag{8}$$

When $\alpha = 1$, $\hat{w}_{1,k}(\boldsymbol{\epsilon}_k; \boldsymbol{\phi}, \boldsymbol{x}) = 1/K$, and it recovers the stochastic gradient VI method [18].

To speed-up learning [17] suggested back-propagating only one sample $\boldsymbol{\epsilon}_j$ with $j \sim p_j = \hat{w}_{\alpha,j}$, which can be easily extended to our framework. Importantly, the use of different $\alpha < 1$ indicates the degree of emphasis placed upon locations where the approximation $q$ under-estimates $p$, and in the extreme case $\alpha \to -\infty$, the algorithm chooses the sample that has the *maximum* unnormalised importance weight. We name this approach *VR-max* and summarise it and the general case in Algorithm 1. Note that VR-max (and VR-$\alpha$ with $\alpha < 0$ and MC approximations) does *not* minimise $D_{1-\alpha}[p||q]$. It is true that $\mathcal{L}_\alpha \geq \log p(\boldsymbol{x})$ for negative $\alpha$ values. However Corollary 1 suggests that the tightest MC approximation for given $K$ has non-positive $\alpha_K$ value, or might not even exist. Furthermore $\alpha_K$ becomes more negative as the mismatch between $q$ and $p$ increases, e.g. VAE uses a uni-modal $q$ distribution to approximate the typically multi-modal exact posterior.

**Algorithm 1** One gradient step for VR-$\alpha$/VR-max with single backward pass. Here $\hat{w}(\epsilon_k; \boldsymbol{x})$ short-hands $\hat{w}_{0,k}(\epsilon_k; \boldsymbol{\phi}, \boldsymbol{x})$ in the main text.

1: given the current datapoint $\boldsymbol{x}$, sample
   $\epsilon_1, ..., \epsilon_K \sim p(\epsilon)$
2: for $k = 1, ..., K$, compute the unnormalised weight

   $\log \hat{w}(\epsilon_k; \boldsymbol{x}) = \log p(g_{\boldsymbol{\phi}}(\epsilon_k), \boldsymbol{x}) - \log q(g_{\boldsymbol{\phi}}(\epsilon_k)|\boldsymbol{x})$

3: choose the sample $\epsilon_j$ to back-propagate:
   if $|\alpha| < \infty$: $j \sim p_k$ where $p_k \propto \hat{w}(\epsilon_k; \boldsymbol{x})^{1-\alpha}$
   if $\alpha = -\infty$: $j = \arg\max_k \log \hat{w}(\epsilon_k; \boldsymbol{x})$
4: return the gradients $\nabla_{\boldsymbol{\phi}} \log \hat{w}(\epsilon_j; \boldsymbol{x})$

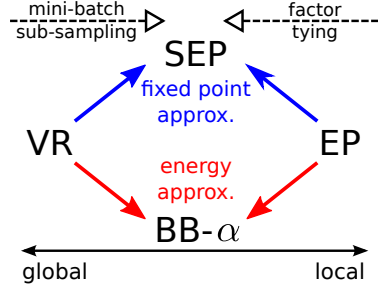

Figure 3: Connecting local and global divergence minimisation.

### 4.3 Stochastic approximation for large-scale learning

VR bounds can also be applied to full Bayesian inference with posterior approximation. However for large datasets full batch learning is very inefficient. Mini-batch training is non-trivial here since the VR bound cannot be represented by the expectation on a datapoint-wise loss, except when $\alpha = 1$. This section introduces two proposals for mini-batch training, and interestingly, this recovers two existing algorithms that were motivated from a different perspective. In the following we define the "average likelihood" $\bar{f}_{\mathcal{D}}(\boldsymbol{\theta}) = [\prod_{n=1}^N p(\boldsymbol{x}_n|\boldsymbol{\theta})]^{\frac{1}{N}}$. Hence the joint distribution can be rewritten as $p(\boldsymbol{\theta}, \mathcal{D}) = p_0(\boldsymbol{\theta}) \bar{f}_{\mathcal{D}}(\boldsymbol{\theta})^N$. Also for a mini-batch of $M$ datapoints $\mathcal{S} = \{\boldsymbol{x}_{n_1}, ..., \boldsymbol{x}_{n_M}\} \sim \mathcal{D}$, we define the "subset average likelihood" $\bar{f}_{\mathcal{S}}(\boldsymbol{\theta}) = [\prod_{m=1}^M p(\boldsymbol{x}_{n_m}|\boldsymbol{\theta})]^{\frac{1}{M}}$.

The first proposal considers *fixed point approximations* with mini-batch sub-sampling. It first derives the fixed point conditions for the variational parameters (e.g. the natural parameters of $q$) using the exact VR bound (4), then design an iterative algorithm using those fixed point equations, but with $\bar{f}_{\mathcal{D}}(\boldsymbol{\theta})$ replaced by $\bar{f}_{\mathcal{S}}(\boldsymbol{\theta})$. The second proposal also applies this subset average likelihood approximation idea, but directly to the VR bound (4) (so this approach is named *energy approximation*):

$$\tilde{\mathcal{L}}_\alpha(q; \mathcal{S}) = \frac{1}{1-\alpha} \log \mathbb{E}_q \left[ \left( \frac{p_0(\boldsymbol{\theta}) \bar{f}_{\mathcal{S}}(\boldsymbol{\theta})^N}{q(\boldsymbol{\theta})} \right)^{1-\alpha} \right]. \tag{9}$$

In the appendix we demonstrate with detailed derivations that fixed point approximation returns Stochastic EP (SEP) [7], and black box alpha (BB-$\alpha$) [16] corresponds to energy approximation. Both algorithms were originally proposed to approximate (power) EP [3, 26], which usually minimises $\alpha$-divergences *locally*, and considers $M = 1$, $\alpha \in [1 - 1/N, 1)$ and exponential family distributions. These approximations were done by factor tying, which significantly reduces the memory overhead of full EP and makes both SEP and BB-$\alpha$ scalable to large datasets just as SVI. The new derivation derivation provides a theoretical justification from energy perspective, and also sheds lights on the connections between *local* and *global* divergence minimisations as depicted in Figure 3. Note that all these methods recover SVI when $\alpha \to 1$, in which global and local divergence minimisation are equivalent. Also these results suggest that recent attempts of distributed posterior approximation (by carving up the dataset into pieces with $M > 1$ [10, 11]) can be extended to both SEP and BB-$\alpha$.

Monte Carlo methods can also be applied to both proposals. For SEP the moment computation can be approximated with MCMC [10, 11]. For BB-$\alpha$ one can show in the same way as to prove Theorem 2 that simple MC approximation in expectation lower-bounds the BB-$\alpha$ energy when $\alpha \leq 1$. In general it is also an open question how to choose $\alpha$ for given the mini-batch size $M$ and the number of samples $K$, but there is evidence that intermediate $\alpha$ values can be superior [27, 28].

## 5 Experiments

We evaluate the VR bound methods on Bayesian neural networks and variational auto-encoders. All the experiments used the ADAM optimizer [29], and the detailed experimental set-up (batch size, learning rate, etc.) can be found in the appendix. The implementation of all the experiments in Python is released at `https://github.com/YingzhenLi/VRbound`.

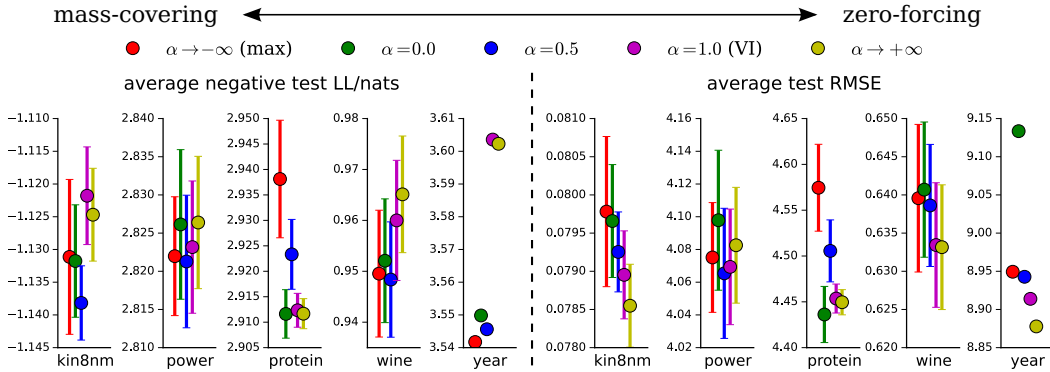

Figure 4: Test LL and RMSE results for Bayesian neural network regression. The lower the better.

## 5.1 Bayesian neural network

The first experiment considers Bayesian neural network regression. The datasets are collected from the UCI dataset repository.[1] The model is a single-layer neural network with 50 hidden units (ReLUs) for all datasets except Protein and Year (100 units). We use a Gaussian prior $\theta \sim \mathcal{N}(\theta; \mathbf{0}, \mathbf{I})$ for the network weights and Gaussian approximation to the true posterior $q(\theta) = \mathcal{N}(\theta; \boldsymbol{\mu}_q, diag(\boldsymbol{\sigma}_q))$. We follow the toy example in Section 3 to consider $\alpha \in \{-\infty, 0.0, 0.5, 1.0, +\infty\}$ in order to examine the effect of mass-covering/zero-forcing behaviour. Stochastic optimisation uses the energy approximation proposed in Section 4.3. MC approximation is also deployed to compute the energy function, in which $K = 100, 10$ is used for small and large datasets (Protein and Year), respectively.

We summarise the test negative log-likelihood (LL) and RMSE with standard error (across different random splits except for Year) for selected datasets in Figure 4, where the full results are provided in the appendix. These results indicate that for posterior approximation problems, the optimal $\alpha$ may vary for different datasets. Also the MC approximation complicates the selection of $\alpha$ (see appendix). Future work should develop algorithms to automatically select the best $\alpha$ values, although a naive approach could use validation sets. We observed two major trends that zero-forcing/mode-seeking methods tend to focus on improving the predictive error, while mass-covering methods returns better calibrated uncertainty estimate and better test log-likelihood. In particular VI returns lower test log-likelihood for most of the datasets. Furthermore, $\alpha = 0.5$ produced overall good results for both test LL and RMSE, possibly because the skew symmetry is centred at $\alpha = 0.5$ and the corresponding divergence is the only symmetric distance measure in the family.

## 5.2 Variational auto-encoder

The second experiments considers variational auto-encoders for unsupervised learning. We mainly compare three approaches: VAE ($\alpha = 1.0$), IWAE ($\alpha = 0$), and VR-max ($\alpha = -\infty$), which are implemented upon the publicly available code.[2] Four datasets are considered: Frey Face (with 10-fold cross validation), Caltech 101 Silhouettes, MNIST and OMNIGLOT. The VAE model has $L = 1, 2$ stochastic layers with deterministic layers stacked between, and the network architecture is detailed in the appendix. We reproduce the IWAE experiments to obtain a fair comparison, since the results in the original publication [17] mismatches those evaluated on the publicly available code.

We report test log-likelihood results in Table 2 by computing $\log p(\boldsymbol{x}) \approx \hat{\mathcal{L}}_{0,5000}(q; \boldsymbol{x})$ following [17]. We also present some samples from the trained models in the appendix. Overall VR-max is almost indistinguishable from IWAE. Other positive alpha settings (e.g. $\alpha = 0.5$) return worse results, e.g. $1374.64 \pm 5.62$ for Frey Face and $-85.50$ for MNIST with $\alpha = 0.5$, $L = 1$ and $K = 5$. These worse results for $\alpha > 0$ indicate the preference of getting tighter approximations to the likelihood function for MLE problems. Small negative $\alpha$ values (e.g. $\alpha = -1.0, -2.0$) returns better results on different splits of the Frey Face data, and overall the best $\alpha$ value is dataset-specific.

Table 2: Average Test log-likelihood. Results for VAE on MNIST and OMNIGLOT are collected from [17].

| Dataset | $L$ | $K$ | VAE | IWAE | VR-max |
|---|---|---|---|---|---|
| Frey Face | 1 | 5 | 1322.96 | **1380.30** | 1377.40 |
| ($\pm$ std. err.) | | | $\pm 10.03$ | $\pm$**4.60** | $\pm 4.59$ |
| Caltech 101 | 1 | 5 | -119.69 | **-117.89** | -118.01 |
| Silhouettes | | 50 | -119.61 | -117.21 | **-117.10** |
| MNIST | 1 | 5 | -86.47 | **-85.41** | -85.42 |
| | | 50 | -86.35 | **-84.80** | -84.81 |
| | 2 | 5 | -85.01 | **-83.92** | -84.04 |
| | | 50 | -84.78 | **-83.05** | -83.44 |
| OMNIGLOT | 1 | 5 | -107.62 | **-106.30** | -106.33 |
| | 1 | 50 | -107.80 | **-104.68** | -105.05 |
| | 2 | 5 | -106.31 | **-104.64** | -104.71 |
| | 2 | 50 | -106.30 | **-103.25** | -103.72 |

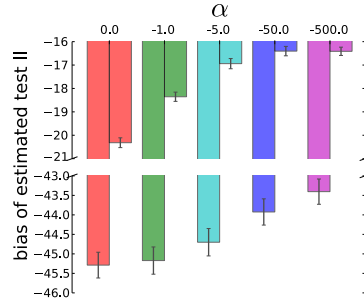

Figure 5: Bias of sampling approximation to. Results for $K = 5, 50$ samples are shown on the left and right, respectively.

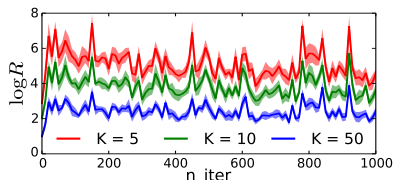

(a) Log of ratio $R = w_{max}/(1 - w_{max})$

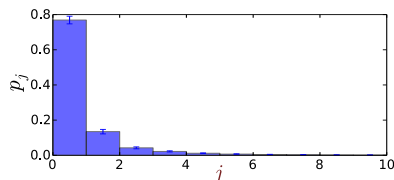

(b) Weights of samples.

Figure 6: Importance weights during training, see main text for details. Best viewed in colour.

VR-max's success might be explained by the tightness of the bound. To evaluate this, we compute the VR bounds on $100$ test datapoints using the 1-layer VAE trained on Frey Face, with $K = \{5, 50\}$ and $\alpha \in \{0, -1, -5, -50, -500\}$. Figure 5 presents the estimated gap $\hat{\mathcal{L}}_{\alpha,K} - \hat{\mathcal{L}}_{0,5000}$. The results indicates that $\hat{\mathcal{L}}_{\alpha,K}$ provides a lower-bound, and that gap is narrowed as $\alpha \to -\infty$. Also increasing $K$ provides improvements. The standard error of estimation is almost constant for different $\alpha$ (with $K$ fixed), and is negligible when compared to the MC approximation bias.

Another explanation for VR-max's success is that, the sample with the largest normalised importance weight $w_{max}$ dominates the contributions of all the gradients. This is confirmed by tracking $R = \frac{w_{max}}{1-w_{max}}$ during training on Frey Face (Figure 6(a)). Also Figure 6(b) shows the 10 largest importance weights from $K = 50$ samples in descending order, which exhibit an exponential decay behaviour, with the largest weight occupying more than $75\%$ of the probability mass. Hence VR-max provides a fast approximation to IWAE when tested on CPUs or multiple GPUs with high communication costs. Indeed our numpy implementation of VR-max achieves up to 3 times speed-up compared to IWAE (9.7s vs. 29.0s per epoch, tested on Frey Face data with $K = 50$ and batch size $M = 100$, CPU info: Intel Core i7-4930K CPU @ 3.40GHz). However this speed advantage is less significant when the gradients can be computed very efficiently on a single GPU.

## 6 Conclusion

We have introduced the variational Rényi bound and an associated optimisation framework. We have shown the richness of the new family, not only by connecting to existing approaches including VI/VB, SEP, BB-$\alpha$, VAE and IWAE, but also by proposing the VR-max algorithm as a new special case. Empirical results on Bayesian neural networks and variational auto-encoders indicate that VR bound methods are widely applicable and can obtain state-of-the-art results. Future work will focus on both experimental and theoretical sides. Theoretical work will study the interaction of the biases introduced by MC approximation and datapoint sub-sampling. A guide on choosing optimal $\alpha$ values are needed for practitioners when applying the framework to their applications.

**Acknowledgements**

We thank the Cambridge MLG members and the reviewers for comments. YL thanks the Schlumberger Foundation FFTF fellowship. RET thanks EPSRC grants # EP/M026957/1 and EP/L000776/1.

## Footnotes

[1]http://archive.ics.uci.edu/ml/datasets.html

[2]https://github.com/yburda/iwae

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
