[Supplementary Material · appendix.pdf]

# Rényi Divergence Variational Inference: Appendix

**Yingzhen Li**
University of Cambridge
Cambridge, CB2 1PZ, UK
yl494@cam.ac.uk

**Richard E. Turner**
University of Cambridge
Cambridge, CB2 1PZ, UK
ret26@cam.ac.uk

The appendix is organised as follows. Section A presents other existing definitions of $\alpha$-divergences. Section B provides the mathematical details for the Bayesian linear regression example. Section C provides the proofs for the main theoretical results. Section D briefly discusses the optimisation issues brought from the selection of $\alpha$ values and the number of MC samples $K$. Section E applies the reparametrization trick to the MC approximated bound, which leads to a unified implementation. Section F demonstrates the connections between the proposed sub-sampling approximation and existing algorithms (SEP [1] and BB-$\alpha$ [2]). Section G provides detailed experimental set-up and further results for the tests considered in the main text.

## A    Other $\alpha$-divergence definitions

Here we include some existing $\alpha$-divergence definitions other than Rényi's.

- Amari's $\alpha$-divergence [3]

$$\mathrm{D}_\alpha[p||q] = \frac{4}{1-\alpha^2} \left( 1 - \int p(\boldsymbol{\theta})^{\frac{1+\alpha}{2}} q(\boldsymbol{\theta})^{\frac{1-\alpha}{2}} d\boldsymbol{\theta} \right).$$

- Tsallis's $\alpha$-divergence [4]

$$\mathrm{D}_\alpha[p||q] = \frac{1}{\alpha-1} \left( \int p(\boldsymbol{\theta})^\alpha q(\boldsymbol{\theta})^{1-\alpha} d\boldsymbol{\theta} - 1 \right).$$

Consider the problem of posterior approximation by minimising an $\alpha$-divergence. When the approximate posterior $q$ has an exponential family form, minimising $\mathrm{D}_\alpha[p||q]$, no matter which definition above is used (although may use different alpha), requires moment matching to the tilted distribution $\tilde{p}_\alpha(\boldsymbol{\theta}) \propto p(\boldsymbol{\theta})^\alpha q(\boldsymbol{\theta})^{1-\alpha}$. In the EP literature Amari's definition is often discussed. We focus on Rényi's definition in the main text simply because $\mathrm{D}_\alpha[q(\boldsymbol{\theta})||p(\boldsymbol{\theta}|\mathcal{D})]$ using Rényi's definition contains $\log p(\mathcal{D})$ that can be cancelled in the same way as VI is derived.

## B    A mean-field approximation example

We present the mean-field approximation method for the VR bound family, with Bayesian linear regression as an illustrating example. Recall the VR bound for $\alpha \neq 1$:

$$\mathcal{L}_\alpha(q; \mathcal{D}) := \frac{1}{1-\alpha} \log \mathbb{E}_q \left[ \left( \frac{p(\boldsymbol{\theta}, \mathcal{D})}{q(\boldsymbol{\theta})} \right)^{1-\alpha} \right], \tag{1}$$

where the $q$ distribution is factorised over the components of $\boldsymbol{\theta} = (\theta_1, ..., \theta_d)$: $q(\boldsymbol{\theta}) = \prod_i q(\theta_i)$. In the following we denote $q_j = q(\theta_j)$ to reduce notational clutter, and re-write the VR bound as

$$\mathcal{L}_\alpha(q; \mathcal{D}) = \frac{1}{1-\alpha} \log \int \prod_i q_i \left( \frac{p(\boldsymbol{\theta}, \mathcal{D})}{\prod_i q_i} \right)^{1-\alpha} d\boldsymbol{\theta}$$

$$= \frac{1}{1-\alpha} \log \int q_j^\alpha \left( \int \prod_{i \neq j} q_i \left( \frac{p(\boldsymbol{\theta}, \mathcal{D})}{\prod_{i \neq j} q_i} \right)^{1-\alpha} d\boldsymbol{\theta}_{i \neq j} \right) d\theta_j$$

$$:= \frac{1}{1-\alpha} \log \int q_j^\alpha \tilde{p}_j^{1-\alpha} d\theta_j + \text{const},$$

where $\tilde{p}_j$ denote the "marginal" distribution satisfying

$$\log \tilde{p}_j = \frac{1}{1-\alpha} \log \int \prod_{i \neq j} q_i \left( \frac{p(\boldsymbol{\theta}, \mathcal{D})}{\prod_{i \neq j} q_i} \right)^{1-\alpha} d\boldsymbol{\theta}_{i \neq j} + \text{const}.$$

Now maximising the VR bound (when $\alpha > 0$, and for $\alpha < 0$ we minimise the bound) is equivalent to minimising $\mathrm{D}_\alpha[q_j||\tilde{p}_j]$ (for $\alpha > 0$, and when $\alpha < 0$ we minimise $\mathrm{D}_{1-\alpha}[\tilde{p}_j||q_j]$), which means $\log q_j = \log \tilde{p}_j + \text{const}$. One can verify that when $\alpha \to 1$ it recovers the traditional variational mean-field approximation

$$\lim_{\alpha \to 1} q_j = \int \prod_{i \neq j} q_i \log p(\boldsymbol{\theta}, \mathcal{D}) d\boldsymbol{\theta}_{i \neq j} + \text{const},$$

and when $\alpha \to 0$ it returns the exact marginal of the posterior distribution $\lim_{\alpha \to 0} q_j = p(\theta_j|\mathcal{D})$.

Now consider Bayesian linear regression with 2-D input $\boldsymbol{x}$ and 1-D output $y$, as an example:

$$\boldsymbol{\theta} \sim \mathcal{N}(\boldsymbol{\theta}; \boldsymbol{\mu}_0, \boldsymbol{\Lambda}_0^{-1}), \quad y|\boldsymbol{x} \sim \mathcal{N}(y; \boldsymbol{\theta}^T \boldsymbol{x}, \sigma^2).$$

Given the observations $\mathcal{D} = \{\boldsymbol{x}_n, y_n\}$, the posterior distribution of $\boldsymbol{\theta}$ can be computed analytically as $p(\boldsymbol{\theta}|\mathcal{D}) = \mathcal{N}(\boldsymbol{\theta}; \boldsymbol{\mu}, \boldsymbol{\Lambda}^{-1})$ with $\boldsymbol{\Lambda} = \boldsymbol{\Lambda}_0 + \frac{1}{\sigma^2} \sum_n \boldsymbol{x}_n \boldsymbol{x}_n^T$ and $\boldsymbol{\Lambda}\boldsymbol{\mu} = \boldsymbol{\Lambda}_0 \boldsymbol{\mu}_0 + \frac{1}{\sigma^2} \sum_n y_n \boldsymbol{x}_n$. To see how the mean-field approach work we explicitly write down the elements of the posterior parameters

$$\boldsymbol{\mu} = \begin{pmatrix} \mu_1 \\ \mu_2 \end{pmatrix}, \quad \boldsymbol{\Lambda} = \begin{pmatrix} \Lambda_{11} & \Lambda_{12} \\ \Lambda_{21} & \Lambda_{22} \end{pmatrix}, \quad \Lambda_{12} = \Lambda_{21},$$

and define $q_i = \mathcal{N}(\theta_i; m_i, \lambda_i^{-1})$ as a univariate Gaussian distribution. Then

$$\log q_1 = \frac{1}{1-\alpha} \log \int q_2(\theta_2) \left( \frac{p(\boldsymbol{\theta}, \mathcal{D})}{q_2(\theta_2)} \right)^{1-\alpha} d\theta_2 + \text{const}$$

$$= \frac{1}{1-\alpha} \log \int \exp \left[ -\frac{1-\alpha}{2} (\boldsymbol{\theta} - \boldsymbol{\mu})^T \boldsymbol{\Lambda} (\boldsymbol{\theta} - \boldsymbol{\mu}) - \frac{\alpha}{2} \lambda_2 (\theta_2 - m_2)^2 \right] d\theta_2 + \text{const}$$

$$= \frac{1}{1-\alpha} \log \int \mathcal{N}(\boldsymbol{\theta}; \boldsymbol{\mu}, \tilde{\boldsymbol{\Sigma}}) d\theta_2 + \text{const}$$

$$= \log \mathcal{N}(\theta_1; m_1, \lambda^{-1}) + \text{const}$$

where the new mean $m_1$ and the precision $\lambda_1$ satisfies

$$m_1 = \mu_1 + C_1(\mu_2 - m_2), \quad C_1 = \frac{\alpha \lambda_2 \Lambda_{12}}{(1-\alpha)|\boldsymbol{\Lambda}| + \alpha \lambda_2 \Lambda_{11}},$$

$$\lambda_1 = \Lambda_{11} - (1-\alpha)\Lambda_{12}((1-\alpha)\Lambda_{22} + \alpha \lambda_2)^{-1} \Lambda_{21}.$$

One can derive the terms $m_2$ and $C_2$ for $q_2$ in the same way, and show that $\boldsymbol{m} = \boldsymbol{\mu}$ is the only stable fixed point of this iterative update. So we have $q_1 = \mathcal{N}(\theta_1; \mu_1, \lambda_1^{-1})$, and similarly $q_2 = \mathcal{N}(\theta_1; \mu_2, \lambda_2^{-1})$ with $\lambda_2 = \Lambda_{22} - (1-\alpha)\Lambda_{21}((1-\alpha)\Lambda_{11} + \alpha \lambda_1)^{-1} \Lambda_{12}$. In this example $\lambda_1, \lambda_2$ are feasible for all $\alpha$, and solving the fixed point equations, finally we have the stable fixed point as

$$\lambda_1 = \rho_\alpha \Lambda_{11}, \quad \lambda_2 = \rho_\alpha \Lambda_{22}, \quad \rho_\alpha = \frac{1}{2\alpha} \left[ (2\alpha - 1) + \sqrt{1 - \frac{4\alpha(1-\alpha)\Lambda_{12}^2}{\Lambda_{11}\Lambda_{22}}} \right].$$

The other solution for the quadratic formula is eliminated since it violates the assumptions that $\lambda_1 > 0$ (when $0 < \alpha < 1$) and $|\mathcal{L}_\alpha| < +\infty$ (when $\alpha < 0$ or $\alpha > 1$, since it requires $|\alpha \text{diag}(\boldsymbol{\lambda}) + (1-\alpha)\boldsymbol{\Lambda}| > 0$). Thus the stable fixed point in this case is unique.

One can show that $\lim_{\alpha \to 1} \lambda_1 = \Lambda_{11}$, $\lim_{\alpha \to 0} \lambda_1 = \Lambda_{11} - \Lambda_{12}\Lambda_{22}^{-1}\Lambda_{21}$ and $\lim_{\alpha \to \pm\infty} \lambda_1 = \Lambda_{11} \pm |\Lambda_{12}|\sqrt{\Lambda_{11}\Lambda_{22}^{-1}}$ (similar results for $\lambda_2$). Also $\rho_\alpha$ is continuous and non-decreasing in $\alpha$. This means one can interpolate between mass-covering and zero-forcing behaviour by increasing $\alpha$ values. Moreover, notice that the limiting case $\alpha \to +\infty$ still returns uncertain estimates, although it is even more over-confident than VI. This is different from maximum a posteriori (MAP) which captures the mode but only returns a point estimate.

## C  Proofs of the main results

We provide the proofs of the theorems presented in section 4 of the main text.

### C.1  Proof of Theorem 2

*Proof.* 1) First we prove for $\alpha \leq 1$, $\mathbb{E}_{\{\boldsymbol{h}_k\}}[\hat{\mathcal{L}}_{\alpha,K}]$ is non-decreasing in $K$. It is straight forward to show the results holds for $\alpha = 1$. We follow the proof in [5] for fixed $\alpha < 1$. Let $K > 1$ and the subset of indices $I = \{i_1, ..., i_{K'}\} \subset \{1, ..., K\}, K' < K$ randomly sampled from integers 1 to $K$. Then for any $\alpha < 1$:

$$
\begin{aligned}
\mathbb{E}_{\{\boldsymbol{h}_k\}_{k=1}^K}[\hat{\mathcal{L}}_{\alpha,K}] &= \frac{1}{1-\alpha}\mathbb{E}_{\{\boldsymbol{h}_k\}}\left[\log \frac{1}{K}\sum_{k=1}^K \left(\frac{p(\boldsymbol{h}_k, \boldsymbol{x})}{q(\boldsymbol{h}_k|\boldsymbol{x})}\right)^{1-\alpha}\right] \\
&= \frac{1}{1-\alpha}\mathbb{E}_{\{\boldsymbol{h}_k\}}\left[\log \mathbb{E}_{I\subset\{1,...,K\}}\left[\frac{1}{K'}\sum_{k=1}^{K'}\left(\frac{p(\boldsymbol{h}_{i_k}, \boldsymbol{x})}{q(\boldsymbol{h}_{i_k})}\right)^{1-\alpha}\right]\right] \\
&\geq \frac{1}{1-\alpha}\mathbb{E}_{\{\boldsymbol{h}_k\}}\left[\mathbb{E}_{I\subset\{1,...,K\}}\left[\log \frac{1}{K'}\sum_{k=1}^{K'}\left(\frac{p(\boldsymbol{h}_{i_k}, \boldsymbol{x})}{q(\boldsymbol{h}_{i_k})}\right)^{1-\alpha}\right]\right] \quad (\log x \text{ is concave})\\
&= \frac{1}{1-\alpha}\mathbb{E}_{\{\boldsymbol{h}_k\}}\left[\log \frac{1}{K'}\sum_{k=1}^{K'}\left(\frac{p(\boldsymbol{h}_k, \boldsymbol{x})}{q(\boldsymbol{h}_k|\boldsymbol{x})}\right)^{1-\alpha}\right] = \mathbb{E}_{\{\boldsymbol{h}_k\}_{k=1}^{K'}}[\hat{\mathcal{L}}_{\alpha,K'}]
\end{aligned}
$$

We used Jensen's inequality of logarithm for the lower-bounding result here. When $\alpha > 1$ we can proof similar result but with inequality reversed, simply because now $1 - \alpha < 0$.

2) Next we prove that, when $K \to \infty$ and $|\mathcal{L}_\alpha| < +\infty$, we have $\mathbb{E}_{\{\boldsymbol{h}_k\}_{k=1}^K}[\hat{\mathcal{L}}_{\alpha,K}] \to \mathcal{L}_\alpha$ if $\hat{\mathcal{L}}_{\alpha,K}$ is absolutely integrable wrt. $qd\mu = dQ$ for all $K \geq 1$ (in other words $\mathbb{E}_{\{\boldsymbol{h}_k\}_{k=1}^K}[|\hat{\mathcal{L}}_{\alpha,K}|] < +\infty$). We only prove it for $\alpha \leq 1$, and for $\alpha > 1$ it can be proved in a similar way. First we use Jensen's inequality again for all finite $K$:

$$
\begin{aligned}
\mathbb{E}_{\{\boldsymbol{h}_k\}_{k=1}^K}[\hat{\mathcal{L}}_{\alpha,K}] &= \frac{1}{1-\alpha}\mathbb{E}_{\{\boldsymbol{h}_k\}}\left[\log \frac{1}{K}\sum_{k=1}^K \left(\frac{p(\boldsymbol{h}_k, \boldsymbol{x})}{q(\boldsymbol{h}_k|\boldsymbol{x})}\right)^{1-\alpha}\right] \\
&\leq \frac{1}{1-\alpha}\log \mathbb{E}_{\{\boldsymbol{h}_k\}}\left[\frac{1}{K}\sum_{k=1}^K \left(\frac{p(\boldsymbol{h}_k, \boldsymbol{x})}{q(\boldsymbol{h}_k|\boldsymbol{x})}\right)^{1-\alpha}\right] = \mathcal{L}_\alpha.
\end{aligned}
$$

This implies $\limsup_{K \to +\infty} \mathbb{E}_{\{\boldsymbol{h}_k\}_{k=1}^K}[\hat{\mathcal{L}}_{\alpha,K}] \leq \mathcal{L}_\alpha$.

Then as an intermediate result we prove $\hat{\mathcal{L}}_{\alpha,K} \to \mathcal{L}_\alpha$ almost surely when $K \to \infty$. For $\alpha \neq 1$, since function $\log$ is continuous we again swap the limit and logarithm:

$$
\lim_{K \to +\infty} \frac{1}{1-\alpha}\log \frac{1}{K}\sum_{k=1}^K \left(\frac{p(\boldsymbol{h}_k, \boldsymbol{x})}{q(\boldsymbol{h}_k|\boldsymbol{x})}\right)^{1-\alpha} = \frac{1}{1-\alpha}\log \lim_{K \to +\infty} \frac{1}{K}\sum_{k=1}^K \left(\frac{p(\boldsymbol{h}_k, \boldsymbol{x})}{q(\boldsymbol{h}_k|\boldsymbol{x})}\right)^{1-\alpha}.
$$

Now since we assume $|\mathcal{L}_\alpha| < +\infty$, this implies $\mathbb{E}_q\left[\left(\frac{p(\boldsymbol{h},\boldsymbol{x})}{q(\boldsymbol{h}|\boldsymbol{x})}\right)^{1-\alpha}\right]$ is finite. Also notice for all $\alpha$ values the ratio $p/q$ is non-negative. Thus by the strong law of large numbers we have

$$\lim_{K\to+\infty}\frac{1}{K}\sum_{k=1}^{K}\left(\frac{p(\boldsymbol{h}_k,\boldsymbol{x})}{q(\boldsymbol{h}_k|\boldsymbol{x})}\right)^{1-\alpha} = \mathbb{E}_{q(\boldsymbol{h}|\boldsymbol{x})}\left[\left(\frac{p(\boldsymbol{h},\boldsymbol{x})}{q(\boldsymbol{h}|\boldsymbol{x})}\right)^{1-\alpha}\right] \text{ a. s.,}$$

then $\hat{\mathcal{L}}_{\alpha,K} \to \mathcal{L}_\alpha$ almost surely as $K \to +\infty$. When $\alpha = 1$ we can use similar method to prove $\lim_{K\to+\infty}\hat{\mathcal{L}}_{1,K} = \mathcal{L}_{\text{VI}}$ almost surely.

Finally, using the non-increasing in $\alpha$ result we will prove later we have $\hat{\mathcal{L}}_{\alpha,K} \geq \hat{\mathcal{L}}_{1,K}$. Thus we can apply Fatou's Lemma and obtain the following almost surely (notice $\mathbb{E}[\hat{\mathcal{L}}_{1,K}] = \mathcal{L}_{\text{VI}}$ for all $K$):

$$\mathcal{L}_\alpha - \mathcal{L}_{\text{VI}} = \mathbb{E}_{\{\boldsymbol{h}_k\}_{k=1}^{K}}\left[\lim_{K\to+\infty}\hat{\mathcal{L}}_{\alpha,K} - \hat{\mathcal{L}}_{1,K}\right]$$

$$\leq \liminf_{K\to+\infty}\mathbb{E}_{\{\boldsymbol{h}_k\}_{k=1}^{K}}[\hat{\mathcal{L}}_{\alpha,K} - \hat{\mathcal{L}}_{1,K}]$$

$$= \liminf_{K\to+\infty}\mathbb{E}_{\{\boldsymbol{h}_k\}_{k=1}^{K}}[\hat{\mathcal{L}}_{\alpha,K}] - \mathcal{L}_{\text{VI}}.$$

Combining with the supremum bound, we have $\mathbb{E}_{\{\boldsymbol{h}_k\}_{k=1}^{K}}[\hat{\mathcal{L}}_{\alpha,K}] \to \mathcal{L}_\alpha$ when $K$ goes to infinity. For $\alpha > 1$ we use Jensen's inequality to bound the limit infimum and the non-increasing property in $\alpha$ to bound the limit supremum. Thus the convergence result holds for all $\alpha \in \{\alpha : |\mathcal{L}_\alpha| < +\infty\}$.

3) $\mathbb{E}[\hat{\mathcal{L}}_{\alpha,K}]$ is non-increasing in $\alpha$: since expectation preserves monotonicity, it is sufficient to prove the result for $\hat{\mathcal{L}}_{\alpha,K}$. This can be proved in similar way as Theorem 3 and 39 in [6], and we include the prove here for completeness. Notice that for $\alpha < \beta$ function $x^{\frac{1-\alpha}{1-\beta}}$ defined on $x > 0$ is convex when $\alpha < 1$ and concave when $\alpha > 1$. So applying Jensen's inequality:

$$\hat{\mathcal{L}}_{\alpha,K} = \frac{1}{1-\alpha}\log\frac{1}{K}\sum_{k=1}^{K}\left(\frac{p(\boldsymbol{h}_k,\boldsymbol{x})}{q(\boldsymbol{h}_k|\boldsymbol{x})}\right)^{1-\alpha} = \frac{1}{1-\alpha}\log\frac{1}{K}\sum_{k=1}^{K}\left(\left(\frac{p(\boldsymbol{h}_k,\boldsymbol{x})}{q(\boldsymbol{h}_k|\boldsymbol{x})}\right)^{1-\beta}\right)^{\frac{1-\alpha}{1-\beta}}$$

$$\geq \frac{1}{1-\alpha}\log\left(\frac{1}{K}\sum_{k=1}^{K}\left(\frac{p(\boldsymbol{h}_k,\boldsymbol{x})}{q(\boldsymbol{h}_k|\boldsymbol{x})}\right)^{1-\beta}\right)^{\frac{1-\alpha}{1-\beta}} = \hat{\mathcal{L}}_{\beta,K}.$$

Continuity in $\alpha$: First we show $\hat{\mathcal{L}}_{\alpha,K}$ is continuous in $\alpha$ when $p(\boldsymbol{h}_k,\boldsymbol{x}) \neq 0$ for $\boldsymbol{h}_k \sim q$. For $\alpha \neq 0, 1, \infty$ and for any sequence $\{\alpha_n\} \to \alpha$ it is sufficient to show that

$$\lim_{n\to\infty}\log\frac{1}{K}\sum_{k}q(\boldsymbol{h}_k|\boldsymbol{x})^{\alpha_n}p(\boldsymbol{h}_k,\boldsymbol{x})^{1-\alpha_n}$$

$$= \log\lim_{n\to\infty}\frac{1}{K}\sum_{k}q(\boldsymbol{h}_k|\boldsymbol{x})^{\alpha_n}p(\boldsymbol{h}_k,\boldsymbol{x})^{1-\alpha_n} \quad (\log x \text{ is a continuous function})$$

$$= \log\frac{1}{K}\sum_{k}\lim_{n\to\infty}q(\boldsymbol{h}_k|\boldsymbol{x})^{\alpha_n}p(\boldsymbol{h}_k,\boldsymbol{x})^{1-\alpha_n} \quad (\text{finite sum})$$

$$= \log\frac{1}{K}\sum_{k}q(\boldsymbol{h}_k|\boldsymbol{x})\left(\frac{p(\boldsymbol{h}_k,\boldsymbol{x})}{q(\boldsymbol{h}_k|\boldsymbol{x})}\right)^{1-\lim_{n\to\infty}\alpha_n} \quad (a^x \text{ is continuous in } x \text{ for all } a > 0)$$

$$= \log\frac{1}{K}\sum_{k}q(\boldsymbol{h}_k|\boldsymbol{x})^{\alpha}p(\boldsymbol{h}_k,\boldsymbol{x})^{1-\alpha}.$$

We note that since we assume $\hat{\mathcal{L}}_{\alpha,K}$ is absolutely integrable, we have $p/q > 0$ almost everywhere on the support of $q$. Hence $\{\hat{\mathcal{L}}_{\alpha_n,K}\}$ has point-wise limit $\hat{\mathcal{L}}_{\alpha,K}$ almost everywhere as $n \to +\infty$.

For $\alpha = 0, 1, \infty$ the Rényi divergence is defined by continuity so one can use the same technique to show the continuity of $\hat{\mathcal{L}}_{\alpha,K}$ on those $\alpha$ values for fixed $K$. Then since $\alpha_n \to \alpha$, for any $\epsilon > 0$, there

exists $n$ that is large enough such that $\alpha_m \in (\alpha - \epsilon, \alpha + \epsilon)$ for all $m > n$. Using the monotonicity result, we have for $\forall m > n$, $\hat{\mathcal{L}}_{\alpha_m, K}$ is bounded in the interval $(\hat{\mathcal{L}}_{\alpha+\epsilon, K}, \hat{\mathcal{L}}_{\alpha-\epsilon, K})$ and by assumption we have $\mathbb{E}[|\hat{\mathcal{L}}_{\alpha-\epsilon, K}|] < +\infty$ and $\mathbb{E}[|\hat{\mathcal{L}}_{\alpha+\epsilon, K}|] < +\infty$. This allows us to apply the dominated convergence theorem to prove $\lim_{n \to +\infty} \mathbb{E}[\hat{\mathcal{L}}_{\alpha_n, K}] = \mathbb{E}[\lim_{n \to +\infty} \hat{\mathcal{L}}_{\alpha_n, K}] = \mathbb{E}[\hat{\mathcal{L}}_{\alpha, K}]$. Thus we have proved that $\mathbb{E}[\hat{\mathcal{L}}_{\alpha, K}]$ is continuous on $\alpha \in \{|\mathcal{L}_\alpha| < +\infty\}$ if $\hat{\mathcal{L}}_{\alpha, K}$ is absolutely integrable.

□

## C.2 Proof of Corollary 1

It is sufficient to prove the corollary for the case $q(\boldsymbol{h}|\boldsymbol{x}) \neq p(\boldsymbol{h}|\boldsymbol{x})$. We first introduce the following lemmas. With overloaded notation, $\mu$ denotes the measure on the corresponding space, which also means $dQ = qd\mu$. As we assume $\mathrm{supp}(p) \subseteq \mathrm{supp}(q)$, there might exist some regions that $q > 0$ but $p = 0$. We define $\rho = Q(\mathrm{supp}(q) \backslash \mathrm{supp}(p))$ and rewrite the computation of $\mathbb{E}[\hat{\mathcal{L}}_{\alpha, K}]$.

**Lemma 1.** *Assume $\rho > 0$. Then for all finite $K$ and $\alpha < 0$, $\mathbb{E}_{\{\boldsymbol{h}_k\}_{k=1}^K}[\hat{\mathcal{L}}_{\alpha, K}(q; \boldsymbol{x})] = -\infty$ and thus $\hat{\mathcal{L}}_{\alpha, K}$ is not integrable wrt. $qd\mu = dQ$.*

*Proof.* We define $\tilde{q}$ as the $q$ distribution restricted on the support of $p$, i.e. $\tilde{q} = q/(1 - \rho)$ defined on $\mathrm{supp}(p)$. Then for any fixed $K < +\infty$ and $\alpha < 0$, we have

$$\mathbb{E}_{\{\boldsymbol{h}_k\}_{k=1}^K \sim q}[\hat{\mathcal{L}}_{\alpha, K}(q; \boldsymbol{x})] = \rho^K \log 0 + \sum_{k=1}^K \binom{K}{k} \rho^{K-k} (1-\rho)^k \left( \mathbb{E}_{\{\boldsymbol{h}_j\}_{j=1}^k \sim \tilde{q}}[\hat{\mathcal{L}}_{\alpha, k}(\tilde{q}; \boldsymbol{x})] + \log k \right)$$
$$- (1 - \rho^K)((1-\alpha)\log(1-\rho) + \log K)$$

Thus $\mathbb{E}_{\{\boldsymbol{h}_k\}_{k=1}^K}[\hat{\mathcal{L}}_{\alpha, K}(q; \boldsymbol{x})] = -\infty$ for all finite $K$ and $\alpha < 0$.                     □

The above example shows the pathology of MC approximation which is further discussed in section D. From now on we assume $\hat{\mathcal{L}}_{\alpha, K}$ is absolutely integrable in order to apply Theorem 2.

**Lemma 2.** *Assume $\alpha < 0$, $\hat{\mathcal{L}}_{\alpha, K}$ absolutely integrable wrt. $qd\mu = dQ$ for all $K$, $\mathcal{L}_\alpha > \mathcal{L}_{VI}$, and $|\mathcal{L}_\alpha| < +\infty$. Then there exists $1 \le K_\alpha < +\infty$ such that for all $K \le K_\alpha < K'$, $\mathbb{E}_{\{\boldsymbol{h}_k\}_{k=1}^K}[\hat{\mathcal{L}}_{\alpha, K}(q; \boldsymbol{x})] \le \log p(\boldsymbol{x}) < \mathbb{E}_{\{\boldsymbol{h}_k\}_{k=1}^{K'}}[\hat{\mathcal{L}}_{\alpha, K'}(q; \boldsymbol{x})]$. Also $K_\alpha$ is **non-decreasing** in $\alpha$ with $\lim_{\alpha \to 0} K_\alpha = +\infty$ and $\lim_{\alpha \to -\infty} K_\alpha \ge 1$.*

*Proof.* 1) Existence of $K_\alpha$: first from Theorem 2 we have $\mathbb{E}[\hat{\mathcal{L}}_{\alpha, K}]$ is non-decreasing in $K$ when $\alpha < 0$. Then since for all $\alpha$, $\mathbb{E}[\hat{\mathcal{L}}_{\alpha, 1}] = \mathcal{L}_{VI} \le \log p(\boldsymbol{x})$, we have $K_\alpha \ge 1$ if $K_\alpha$ exists. Also from Theorem 2 we have $\lim_{K \to +\infty} \mathbb{E}[\hat{\mathcal{L}}_{\alpha, K}] = \mathcal{L}_\alpha > \log p(\boldsymbol{x})$ for all $\alpha < 0$. Hence for $\epsilon = \mathcal{L}_\alpha - \log p(\boldsymbol{x})$ there exist $K$ that is finite but large enough such that $\mathcal{L}_\alpha - \mathbb{E}[\hat{\mathcal{L}}_{\alpha, K'}] < \epsilon$ for all $K' > K$. Now we can define $\epsilon = \mathcal{L}_\alpha - \mathcal{L}_{VI}$ and take $K_\alpha$ as the minimum of such $K$, and it is straight-forward to show that $1 \le K_\alpha < +\infty$.

2) $K_\alpha$ is non-decreasing in $\alpha$: suppose there exist $\alpha > \beta$ such that $K_\alpha < K_\beta$. Then there exist $K_\alpha < K \le K_\beta$ such that $\mathbb{E}[\hat{\mathcal{L}}_{\alpha, K}] > \log p(\boldsymbol{x}) \ge \mathbb{E}[\hat{\mathcal{L}}_{\beta, K}]$. But Theorem 2 says $\mathbb{E}[\hat{\mathcal{L}}_{\alpha, K}]$ is non-increasing in $\alpha$, a contradiction.

3) Since $\lim_{K \to +\infty} \mathbb{E}[\hat{\mathcal{L}}_{\alpha, K}] = \mathcal{L}_\alpha$ and $\mathcal{L}_\alpha \downarrow \log p(\boldsymbol{x})$ when $\alpha \uparrow 0$, we have $\lim_{\alpha \to 0} K_\alpha = +\infty$. Also since $K_\alpha$ is non-decreasing in $\alpha$ and is lower-bounded by 1, we have the limit exists and $\lim_{\alpha \to -\infty} K_\alpha \ge 1$.                     □

Now we prove Corollary 1, and we only prove it with the conditions assumed in Lemma 2 since $K_\alpha = +\infty$ for the other cases, and if so for all $\alpha < 0$, then $\alpha_K = -\infty$ for all finite $K$.

*Proof.* 1) Existence of $\alpha_K$ for $\lim_{\alpha \to -\infty} K_\alpha < K < +\infty$: from Lemma 2 we can find $\alpha > \beta$ such that $K_\alpha \ge K \ge K_\beta$. This means $\mathbb{E}[\hat{\mathcal{L}}_{\alpha, K}] \le \log p(\boldsymbol{x}) \le \mathbb{E}[\hat{\mathcal{L}}_{\beta, K}]$. Since $\mathbb{E}[\hat{\mathcal{L}}_{\alpha, K}]$ is continuous in $\alpha$ for any fixed $K$, there exits $\alpha \le \gamma \le \beta$ to have $\mathbb{E}[\hat{\mathcal{L}}_{\gamma, K}] = \log p(\boldsymbol{x})$. Note that $\gamma$ might not be

unique, so we define $\alpha_K$ as the minimum of such $\gamma$, which also gives $\mathbb{E}[\hat{\mathcal{L}}_{\alpha,K}] > \log p(\boldsymbol{x})$ for all $\alpha < \alpha_K$.

2) $\alpha_K$ is non-decreasing in $K$: suppose there exist $K < K'$ with $\alpha_K > \alpha_{K'}$. Then we can find $\alpha_K > \alpha > \alpha_{K'}$ such that $\mathbb{E}[\hat{\mathcal{L}}_{\alpha,K}] > \log p(\boldsymbol{x}) = \mathbb{E}[\hat{\mathcal{L}}_{\alpha_{K'},K'}] \geq \mathbb{E}[\hat{\mathcal{L}}_{\alpha,K'}]$. But from Theorem 2 $\mathbb{E}[\hat{\mathcal{L}}_{\alpha,K}]$ is non-decreasing in $K$, a contradiction.

3) Since $\lim_{K \to +\infty} \mathbb{E}[\hat{\mathcal{L}}_{\alpha,K}] = \mathcal{L}_\alpha$ and $\mathcal{L}_\alpha \downarrow \log p(\boldsymbol{x})$ when $\alpha \uparrow 0$, we have $\lim_{K \to +\infty} \alpha_K = 0$. Also for all $\alpha$, $\mathbb{E}[\hat{\mathcal{L}}_{\alpha,1}] = \mathcal{L}_{VI} \leq \log p(\boldsymbol{x})$, so $\lim_{K \to 1} \alpha_K = -\infty$.

$\square$

## D   Optimisation issues with $\alpha$-divergences and MC approximations

It is in general an outstanding research question on how to select the divergence measure for a particular machine learning problem. In our case this corresponds to selecting the $\alpha$ value. Also an approximate inference algorithm can be evaluated with different performance measures, and it is impossible to find a single algorithm value that returns the best performance on all evaluations. Thus we only present the evaluation in test error and test log-likelihood in the main text.

We discuss two conjectures to explain the difficulty of selecting $\alpha$ in the Bayesian neural network experiments. The first conjecture is that zero-forcing algorithms tend to favour minimising the test error, while mass-covering methods tend to improve the test log-likelihood. However zero-forcing methods can fail as it might miss an important mode due to local optima. Similarly mass-covering methods can be pathological if the exact posterior includes modes that are very far away from each other. Furthermore, the form of the posterior will change with the number of observed datapoints $N$, so the "optimal" setting of $\alpha$ for a fixed task may change with $N$.

The second conjecture states that the MC approximation complicates the selection of $\alpha$, since it favours zero-forcing (because of the bias introduced). For example, in order to maximize the quantity of the MC approximation the algorithm need to make $\mathbb{E}[\hat{\mathcal{L}}_{\alpha,K}]$ finite first. However, Lemma 1 indicates that, if $\rho > 0$, then for finite sample size, there's a small probability $\rho^K$ that the MC approximation goes wrong. Hence to avoid this pathology the optimisation procedure will ensure $q = 0$ whenever $p$ is zero. Combining with Theorem 2, we conjecture that the MC approximation makes the algorithm more "VI-like" compared to the exact case. In other words, when MC approximation is deployed, the effective $\alpha$ value is closer to $\alpha = 1$ which is the value for VI (consider $K = 1$). This means, if there exists $\alpha_{\mathrm{opt}} \neq 1$ for a specific task, in practice one should use $\alpha \leq \alpha_{\mathrm{opt}}$ (for $\alpha_{\mathrm{opt}} < 1$, and should use $\alpha \geq \alpha_{\mathrm{opt}}$ if $\alpha_{\mathrm{opt}} > 1$) when running the MC algorithm. In general one should be very careful when estimating the ratio between distribution with Monte Carlo methods. Also the introduced MC approach usually has higher variance compared to the variational case, so further control variate techniques should be applied to reduce the sampling variance.

Still we want to emphasize that for many problems, minimising an $\alpha$-divergence other than the KL-divergence can be very useful, even when with MC approximations. Approximate EP has been applied to deep Gaussian process regression and has shown to achieve the state-of-the-art results for benchmark datasets [7]. A recent paper [8] tested BB-$\alpha$ for model-based reinforcement learning with Bayesian neural networks. In their tests using $\alpha = 0.5$ successfully captured the bi-modality and heteroskedasticity in the predictive distribution, while VI failed disastrously.

## E   Unified implementation: derivation details

We provide detailed derivations of the gradient computation here. Recall from the main text that when $\alpha \neq 1$, the VR bound with the reparameterization trick becomes

$$\mathcal{L}_\alpha(q_\phi; \boldsymbol{x}) = \frac{1}{1-\alpha} \log \mathbb{E}_\epsilon \left[ \left( \frac{p(g_\phi(\boldsymbol{\epsilon}), \boldsymbol{x})}{q(g_\phi(\boldsymbol{\epsilon}))} \right)^{1-\alpha} \right]. \tag{2}$$

So the distribution $p(\boldsymbol{\epsilon})$ does not depend on the recognition model. We short-hand $g_{\boldsymbol{\phi}} = g_{\boldsymbol{\phi}}(\boldsymbol{\epsilon})$, then,

$$
\begin{aligned}
\nabla_{\boldsymbol{\phi}} \mathcal{L}_{\alpha}(q_{\boldsymbol{\phi}}; \boldsymbol{x}) &= \frac{1}{1-\alpha} \nabla_{\boldsymbol{\phi}} \log \mathbb{E}_{\boldsymbol{\epsilon}} \left[ \left( \frac{p(g_{\boldsymbol{\phi}}, \boldsymbol{x})}{q(g_{\boldsymbol{\phi}})} \right)^{1-\alpha} \right] \\
&= \frac{1}{1-\alpha} \left( \mathbb{E}_{\boldsymbol{\epsilon}} \left[ \left( \frac{p(g_{\boldsymbol{\phi}}, \boldsymbol{x})}{q(g_{\boldsymbol{\phi}})} \right)^{1-\alpha} \right] \right)^{-1} \mathbb{E}_{\boldsymbol{\epsilon}} \left[ \nabla_{\boldsymbol{\phi}} \left( \frac{p(g_{\boldsymbol{\phi}}, \boldsymbol{x})}{q(g_{\boldsymbol{\phi}})} \right)^{1-\alpha} \right] \\
&= \frac{1}{1-\alpha} \left( \mathbb{E}_{\boldsymbol{\epsilon}} \left[ \left( \frac{p(g_{\boldsymbol{\phi}}, \boldsymbol{x})}{q(g_{\boldsymbol{\phi}})} \right)^{1-\alpha} \right] \right)^{-1} \mathbb{E}_{\boldsymbol{\epsilon}} \left[ \left( \frac{p(g_{\boldsymbol{\phi}}, \boldsymbol{x})}{q(g_{\boldsymbol{\phi}})} \right)^{1-\alpha} \nabla_{\boldsymbol{\phi}}(1-\alpha) \log \frac{p(g_{\boldsymbol{\phi}}, \boldsymbol{x})}{q(g_{\boldsymbol{\phi}})} \right] \\
&= \mathbb{E}_{\boldsymbol{\epsilon}} \left[ w_{\alpha}(\boldsymbol{\epsilon}; \boldsymbol{\phi}, \boldsymbol{x}) \nabla_{\boldsymbol{\phi}} \log \frac{p(g_{\boldsymbol{\phi}}, \boldsymbol{x})}{q(g_{\boldsymbol{\phi}})} \right].
\end{aligned}
$$

Here we define

$$
w_{\alpha}(\boldsymbol{\epsilon}; \boldsymbol{\phi}, \boldsymbol{x}) := \left( \frac{p(g_{\boldsymbol{\phi}}, \boldsymbol{x})}{q(g_{\boldsymbol{\phi}})} \right)^{1-\alpha} \Bigg/ \mathbb{E}_{\boldsymbol{\epsilon}} \left[ \left( \frac{p(g_{\boldsymbol{\phi}}, \boldsymbol{x})}{q(g_{\boldsymbol{\phi}})} \right)^{1-\alpha} \right]. \tag{3}
$$

For MC approximation with finite $K$ samples, one can use the same technique to show that

$$
\nabla_{\boldsymbol{\phi}} \hat{\mathcal{L}}_{\alpha, K}(q_{\boldsymbol{\phi}}; \boldsymbol{x}) = \sum_{k=1}^{K} \left[ \hat{w}_{\alpha, k}(\boldsymbol{\epsilon}_k; \boldsymbol{\phi}, \boldsymbol{x}) \nabla_{\boldsymbol{\phi}} \log \frac{p(g_{\boldsymbol{\phi}}(\boldsymbol{\epsilon}_k), \boldsymbol{x})}{q(g_{\boldsymbol{\phi}}(\boldsymbol{\epsilon}_k))} \right].
$$

with the importance weights

$$
\hat{w}_{\alpha, k}(\boldsymbol{\epsilon}_k; \boldsymbol{\phi}, \boldsymbol{x}) := \left( \frac{p(g_{\boldsymbol{\phi}}(\boldsymbol{\epsilon}_k), \boldsymbol{x})}{q(g_{\boldsymbol{\phi}}(\boldsymbol{\epsilon}_k))} \right)^{1-\alpha} \Bigg/ \sum_{k=1}^{K} \left( \frac{p(g_{\boldsymbol{\phi}}(\boldsymbol{\epsilon}_k), \boldsymbol{x})}{q(g_{\boldsymbol{\phi}}(\boldsymbol{\epsilon}_k))} \right)^{1-\alpha}. \tag{4}
$$

One can show that $\lim_{\alpha \to 1} w_{\alpha}(\boldsymbol{\epsilon}; \boldsymbol{\phi}, \boldsymbol{x}) = 1$ and $\lim_{\alpha \to 1} \hat{w}_{\alpha, k}(\boldsymbol{\epsilon}_k; \boldsymbol{\phi}, \boldsymbol{x}) = 1/K$. This indicates the recovery of the original VAE algorithm.

# F  Stochastic approximation for large-scale learning: derivations

This section shows the connection between VR bound optimisation and the recently proposed algorithms: SEP [1] and BB-$\alpha$ [2], by taking $M = 1$ and $\alpha = 1 - \beta/N$.

Recall that in the main text we define the "average likelihood" $\bar{f}_{\mathcal{D}}(\boldsymbol{\theta}) = [\prod_{n=1}^{N} p(\boldsymbol{x}_n | \boldsymbol{\theta})]^{\frac{1}{N}}$. Hence the joint distribution can be rewritten as $p(\boldsymbol{\theta}, \mathcal{D}) = p_0(\boldsymbol{\theta}) \bar{f}_{\mathcal{D}}(\boldsymbol{\theta})^N$. Also for a mini-batch of $M$ datapoints $\mathcal{S} = \{\boldsymbol{x}_{n_1}, ..., \boldsymbol{x}_{n_m}\} \sim \mathcal{D}$, we define the "subset average likelihood" $\bar{f}_{\mathcal{S}} = [\prod_{m=1}^{M} p(\boldsymbol{x}_{n_m} | \boldsymbol{\theta})]^{\frac{1}{M}}$. When $M = 1$ we also write $\bar{f}_{\mathcal{S}}(\boldsymbol{\theta}) = f_n(\boldsymbol{\theta})$ for $\mathcal{S} = \{\boldsymbol{x}_n\}$.

Now assume the posterior approximation is defined as $q(\boldsymbol{\theta}) = \frac{1}{Z_q} p_0(\boldsymbol{\theta}) t(\boldsymbol{\theta})^N$. Often $t(\boldsymbol{\theta})$ is chose to have an exponential family form $t(\boldsymbol{\theta}) \propto \exp[\langle \boldsymbol{\lambda}, \boldsymbol{\Phi}(\boldsymbol{\theta}) \rangle]$ with $\boldsymbol{\Phi}(\boldsymbol{\theta})$ denoting the sufficient statistic. Then picking $\alpha = 1 - \beta/N$, $\beta \neq 0$, we have the exact VR bound as

$$
\mathcal{L}_{\alpha}(q; \mathcal{D}) = \log Z_q + \frac{N}{\beta} \log \mathbb{E}_q \left[ \left( \frac{\bar{f}_{\mathcal{D}}(\boldsymbol{\theta})}{t(\boldsymbol{\theta})} \right)^{\beta} \right] \tag{5}
$$

The first proposal considers deriving the exact fixed point conditions, then approximating them with mini-batch sub-sampling. In our example the exact fixed point condition for the variational parameters $\boldsymbol{\lambda}$ is

$$
\nabla_{\boldsymbol{\lambda}} \mathcal{L}_{\alpha}(q; \mathcal{D}) = 0 \quad \Rightarrow \quad \mathbb{E}_q[\boldsymbol{\Phi}(\boldsymbol{\theta})] = \mathbb{E}_{\tilde{p}_{\alpha}}[\boldsymbol{\Phi}(\boldsymbol{\theta})], \tag{6}
$$

with the tilted distribution defined as

$$
\tilde{p}_{\alpha}(\boldsymbol{\theta}) \propto q(\boldsymbol{\theta})^{\alpha} p_0(\boldsymbol{\theta})^{1-\alpha} \bar{f}_{\mathcal{D}}(\boldsymbol{\theta})^{N(1-\alpha)} \propto p_0(\boldsymbol{\theta}) t(\boldsymbol{\theta})^{N-\beta} \bar{f}_{\mathcal{D}}(\boldsymbol{\theta})^{\beta}.
$$

Now given a mini-batch of datapoints $\mathcal{S}$, the moment matching update can be approximated by replacing $\bar{f}_{\mathcal{D}}(\boldsymbol{\theta})$ with $\bar{f}_{\mathcal{S}}(\boldsymbol{\theta}) = [\prod_{m=1}^{M} p(\boldsymbol{x}_{n_m} | \boldsymbol{\theta})]^{\frac{1}{M}}$. More precisely, each iteration we sample a

subset of data $\mathcal{S} = \{\boldsymbol{x}_{n_1}, ..., \boldsymbol{x}_{n_M}\} \sim \mathcal{D}$, and compute the new update for $\boldsymbol{\lambda}$ by first computing $\tilde{p}_{\alpha,\mathcal{S}}(\boldsymbol{\theta}) \propto p_0(\boldsymbol{\theta})t(\boldsymbol{\theta})^{N-\beta}\bar{f}_{\mathcal{S}}(\boldsymbol{\theta})^\beta$ then taking $\mathbb{E}_q[\boldsymbol{\Phi}(\boldsymbol{\theta})] \leftarrow \mathbb{E}_{\tilde{p}_{\alpha,\mathcal{S}}}[\boldsymbol{\Phi}(\boldsymbol{\theta})]$. This method returns SEP when $M = 1$, i.e. in each iteration only one datapoint is sampled to update the approximate posterior.

The second proposal also applies this subset average likelihood approximation idea, but directly to the VR bound (5), with $\mathbb{E}_\mathcal{S}$ denotes the expectation over mini-batch sub-sampling:

$$\mathbb{E}_\mathcal{S}\left[\tilde{\mathcal{L}}_\alpha(q;\mathcal{S})\right] = \log Z_q + \frac{N}{\beta}\mathbb{E}_\mathcal{S}\left[\log\mathbb{E}_q\left[\left(\frac{\bar{f}_\mathcal{S}(\boldsymbol{\theta})}{t(\boldsymbol{\theta})}\right)^\beta\right]\right]. \tag{7}$$

It recovers the energy function of BB-$\alpha$ when $M = 1$. Note that the original paper [2] uses an adapted form of Amari's $\alpha$-divergence, and the $\alpha$ value in the BB-$\alpha$ algorithm corresponds to $\beta$ in our exposition. Now the gradient of this approximated energy function becomes

$$\nabla_{\boldsymbol{\lambda}}\mathbb{E}_\mathcal{S}\left[\tilde{\mathcal{L}}_\alpha(q;\mathcal{S})\right] = N(\mathbb{E}_q[\boldsymbol{\Phi}(\boldsymbol{\theta})] - \mathbb{E}_\mathcal{S}\mathbb{E}_{\tilde{p}_{\alpha,\mathcal{S}}}[\boldsymbol{\Phi}(\boldsymbol{\theta})]). \tag{8}$$

Both SEP and BB-$\alpha$ return SVI when $\alpha \to 1$ (or equivalently $\beta \to 0$). But for other $\alpha$ values it is important to note that these two proposals return different optimum at convergence. BB-$\alpha$ requires averages the moment of the tilted distribution $\mathbb{E}_\mathcal{S}\mathbb{E}_{\tilde{p}_{\alpha,\mathcal{S}}}[\boldsymbol{\Phi}(\boldsymbol{\theta})]$. However SEP first compute the inverse mapping from the moment $\mathbb{E}_{\tilde{p}_{\alpha,\mathcal{S}}}[\boldsymbol{\Phi}(\boldsymbol{\theta})]$ to obtain the natural parameters $\boldsymbol{\lambda}_\mathcal{S}$, then update the $q$ distribution by $\boldsymbol{\lambda} \leftarrow \mathbb{E}_\mathcal{S}[\boldsymbol{\lambda}_\mathcal{S}]$. In general the inverse mapping is non-linear so the fixed point conditions of SEP and BB-$\alpha$ are different.

SEP is arguably more well justified since it returns the exact posterior if the approximation family $\mathcal{Q}$ is large enough to include the correct solution, just like VI and VR computed on the whole dataset. BB-$\alpha$ might still be biased even in this scenario. But BB-$\alpha$ is much simpler to implement since the energy function can be optimised with stochastic gradient descent. Indeed the authors of [2] considered the same black-box approach as to VI, by computing a stochastic estimate of the energy function then using automatic differentiation tools to obtain the gradients.

We also provide a bound of the energy approximation (7) by the following theorem.

**Theorem 3.** *If the approximate distribution $q(\boldsymbol{\theta})$ is Gaussian $\mathcal{N}(\boldsymbol{\mu}, \boldsymbol{\Sigma})$, and the likelihood functions has an exponential family form $p(\boldsymbol{x}|\boldsymbol{\theta}) = \exp[\langle\boldsymbol{\theta}, \boldsymbol{\Psi}(\boldsymbol{x})\rangle - A(\boldsymbol{\theta})]$, then for $\alpha \leq 1$ and $r > 1$ the stochastic approximation is bounded by*

$$\mathbb{E}_\mathcal{S}[\tilde{\mathcal{L}}_\alpha(q;\mathcal{S})] \leq \mathcal{L}_{1-(1-\alpha)r}(q;\mathcal{D}) + \frac{N^2(1-\alpha)r}{2(r-1)}\mathrm{tr}(\boldsymbol{\Sigma}\mathrm{Cov}_{\mathcal{S}\sim\mathcal{D}}(\bar{\boldsymbol{\Psi}}_\mathcal{S})).$$

*Proof.* We substitute the exponential family likelihood term into the stochastic approximation of the VR bound with $\alpha < 1$, and use Hölder's inequality for any $1/r + 1/s = 1$, $r > 1$ (define $\tilde{\alpha} = 1 - (1-\alpha)r$):

$$\mathbb{E}_\mathcal{S}[\tilde{\mathcal{L}}_\alpha(q;\mathcal{S})] = \frac{1}{1-\alpha}\log\mathbb{E}_q\left[\left(\frac{p_0(\boldsymbol{\theta})\bar{f}_\mathcal{D}(\boldsymbol{\theta})^N}{q(\boldsymbol{\theta})}\frac{\bar{f}_\mathcal{S}(\boldsymbol{\theta})^N}{\bar{f}_\mathcal{D}(\boldsymbol{\theta})^N}\right)^{1-\alpha}\right]$$

$$\leq \mathcal{L}_{\tilde{\alpha}}(q;\mathcal{D}) + \frac{1}{(1-\alpha)s}\mathbb{E}_\mathcal{S}\left\{\log\mathbb{E}_q[\exp[N(1-\alpha)s\langle\bar{\boldsymbol{\Psi}}_\mathcal{S} - \bar{\boldsymbol{\Psi}}_\mathcal{D}, \boldsymbol{\theta}\rangle]]\right\}$$

$$= \mathcal{L}_{\tilde{\alpha}}(q;\mathcal{D}) + \frac{1}{(1-\alpha)s}\mathbb{E}_\mathcal{S}[K_{\boldsymbol{\theta}}(N(1-\alpha)s(\bar{\boldsymbol{\Psi}}_\mathcal{S} - \bar{\boldsymbol{\Psi}}_\mathcal{D}))],$$

where $\bar{\boldsymbol{\Psi}}_\mathcal{S}$ and $\bar{\boldsymbol{\Psi}}_\mathcal{D}$ denote the mean of the sufficient statistic $\boldsymbol{\Psi}(\boldsymbol{x})$ on the mini-batch $\mathcal{S}$ and the whole dataset $\mathcal{D}$, respectively. For Gaussian distribution $q(\boldsymbol{\theta}) = \mathcal{N}(\boldsymbol{\mu}, \boldsymbol{\Sigma})$ the cumulant generating function $K_{\boldsymbol{\theta}}(\boldsymbol{t})$ has a closed form

$$K_{\boldsymbol{\theta}}(\boldsymbol{t}) = \boldsymbol{\mu}^T\boldsymbol{t} + \frac{1}{2}\boldsymbol{t}^T\boldsymbol{\Sigma}\boldsymbol{t}.$$

Define $\boldsymbol{t}_{\mathcal{S}} = N(1-\alpha)s\Delta_{\mathcal{S}}$ with $\Delta_{\mathcal{S}} = \bar{\boldsymbol{\Psi}}_{\mathcal{S}} - \bar{\boldsymbol{\Psi}}_{\mathcal{D}}$, then $\mathbb{E}_{\mathcal{S}}[\boldsymbol{t}_{\mathcal{S}}] = \boldsymbol{0}$ and the upper-bound becomes

$$\mathbb{E}_{\mathcal{S}}[\tilde{\mathcal{L}}_{\alpha}(q;\mathcal{S})] \leq \mathcal{L}_{\tilde{\alpha}}(q;\mathcal{D}) + \frac{1}{(1-\alpha)s}\mathbb{E}_{\mathcal{S}}[K_{\boldsymbol{\theta}}(\boldsymbol{t}_{\mathcal{S}})]$$

$$= \mathcal{L}_{\tilde{\alpha}}(q;\mathcal{D}) + \frac{1}{(1-\alpha)s}\mathbb{E}_{\mathcal{S}}[\boldsymbol{\mu}^T\boldsymbol{t}_{\mathcal{S}} + \frac{1}{2}\boldsymbol{t}_{\mathcal{S}}^T\boldsymbol{\Sigma}\boldsymbol{t}_{\mathcal{S}}]$$

$$= \mathcal{L}_{\tilde{\alpha}}(q;\mathcal{D}) + \frac{N^2(1-\alpha)s}{2}\mathbb{E}_{\mathcal{S}}[\Delta_{\mathcal{S}}^T\boldsymbol{\Sigma}\Delta_{\mathcal{S}}]$$

$$= \mathcal{L}_{\tilde{\alpha}}(q;\mathcal{D}) + \frac{N^2(1-\alpha)s}{2}\text{tr}(\boldsymbol{\Sigma}\text{Cov}_{\mathcal{S}\sim\mathcal{D}}(\bar{\boldsymbol{\Psi}}_{\mathcal{S}})).$$

Applying the condition of Hölder's inequality $1/r + 1/s = 1$ proves the result. $\qquad\square$

The following corollary is a direct result of Theorem 3 applied to BB-$\alpha$. Note here we follow the convention of the original paper [2] to use $M = 1$ and overload the notation $\alpha = \beta$ and $\mathcal{L}_{BB-\alpha}(q;\mathcal{D}) = \mathbb{E}_{\{\boldsymbol{x}_n\}}\left[\tilde{\mathcal{L}}_{1-\alpha/N}(q;\{\boldsymbol{x}_n\})\right]$.

**Corollary 2.** *Assume the approximate posterior and the likelihood functions satisfy the assumptions in Theorem 3, then for $\alpha > 0$ and $r > 1$, the black-box alpha energy function is upper-bounded by*

$$\mathcal{L}_{BB-\alpha}(q;\mathcal{D}) \leq \mathcal{L}_{1-\frac{\alpha r}{N}}(q;\mathcal{D}) + \frac{N\alpha r}{2(r-1)}\text{tr}(\boldsymbol{\Sigma}\text{Cov}_{\mathcal{D}}(\boldsymbol{\Psi})).$$

## G Further experimental details and results

### G.1 Bayesian neural network

We detail the experimental set-up of the Bayesian neural network example. For regression tests, we consider Protein and Year as the large datasets and the others as small datasets. The likelihood function is defined as $p(y|\boldsymbol{x},\boldsymbol{\theta}) = \mathcal{N}(y; F_{\boldsymbol{\theta}}(\boldsymbol{x}), \sigma^2)$ where $F_{\boldsymbol{\theta}}(\boldsymbol{x})$ denotes the non-linear transform from the neural network with weights $\boldsymbol{\theta}$. We use unit Gaussian prior $\boldsymbol{\theta} \sim \mathcal{N}(\boldsymbol{\theta}; \boldsymbol{0}, \boldsymbol{I})$ and Gaussian approximation $q(\boldsymbol{\theta}) = \mathcal{N}(\boldsymbol{\theta}; \boldsymbol{\mu}_q, diag(\boldsymbol{\sigma}_q))$, where we fit the parameters of $q$ and the noise level $\sigma$ by optimising the lower-bound. For all datasets we use single-layer neural networks with 50 hidden units (ReLUs) for datasets except Protein and Year (100 units). The methods for comparison were run for 500 epochs on the small datasets and 100, 40 epochs for the large datasets Protein and Year, respectively. We used ADAM [9] for optimisation with learning rate 0.001 and the standard setting for other parameters. For stochastic optimisation we used learning rate 0.001, mini-batch size $M = 32$ and number of samples $K = 100, 10$ for small and large datasets. The number of dataset random splits is 20 except for the large datasets, which is 5 and 1 for Protein and Year, respectively.

The full test results are provided in Figure 1 and Table 1, 2. In the tables the best performing results are underlined, while the worse cases are also bold-faced. Clearly the optimal $\alpha$ setting is dataset dependent, although for Boston and Power the performances are very similar. Also for Naval mass-covering seems to be harmful not only for predictive error but also for test log-likelihood measure. Overall mode-seeking methods tend to focus on improving the predictive error, while mass-covering regimes often return better test log-likelihood.

### G.2 Variational auto-encoder

We describe the network architecture tested in the VAE experiments. The number of stochastic layers $L$, number of hidden units, and the activation function are summarised in Table 3. The prefix of the number indicates whether this layer is **d**eterministic or **s**tochastic, e.g. d500-s200 stands for a neural network with one deterministic layer of 500 units followed by a stochastic layer of 200 units. For Frey Face data we train the models using learning rate 0.0005 and mini-batch size 100. For MNIST and OMNIGLOT we reuse the settings from [5]: the training process runs for $3^i$ passes with learning rate $0.0001 \cdot 10^{-i/7}$ for $i = 0, ..., 7$, and the batch size is 20. For Caltech Silhouettes we use the same settings as MNIST and OMNIGLOT except that the training proceeded for $\sum_{i=0}^{7} 2^i = 255$ epochs.

We also present some samples from the VR-max trained auto-encoders in Figure 2, and note that the visual quality of these samples are almost identical to those from IWAE.

Figure 1: Test LL and RMSE results for Bayesian neural network regression. The lower the better.

Table 1: Regression experiment: Average negative test log likelihood/nats

| Dataset | N | D | $\alpha \to -\infty$ | $\alpha = 0.0$ | $\alpha = 0.5$ | $\alpha = 1.0$ (**VI**) | $\alpha \to +\infty$ |
|---|---|---|---|---|---|---|---|
| boston | 506 | 13 | 2.47±0.08 | 2.47±0.07 | _2.46±0.07_ | **2.52±0.03** | 2.50±0.05 |
| concrete | 1030 | 8 | 3.09±0.02 | _3.08±0.02_ | 3.09±0.02 | 3.11±0.02 | **3.12±0.02** |
| energy | 768 | 8 | 1.39±0.02 | **1.42±0.02** | 1.40±0.03 | _0.77±0.02_ | 1.23±0.01 |
| naval | 11934 | 16 | -3.43±0.08 | **-3.02±0.48** | -3.58±0.08 | _-6.49±0.04_ | -6.47±0.09 |
| kin8nm | 8192 | 8 | -1.13±0.01 | -1.13±0.01 | _-1.14±0.01_ | **-1.12±0.01** | -1.12±0.01 |
| power | 9568 | 4 | 2.82±0.01 | 2.83±0.01 | _2.82±0.01_ | 2.82±0.01 | **2.83±0.01** |
| protein | 45730 | 9 | **2.94±0.01** | _2.91±0.00_ | 2.92±0.01 | 2.91±0.00 | 2.91±0.00 |
| wine | 1588 | 11 | 0.95±0.01 | 0.95±0.01 | _0.95±0.01_ | 0.96±0.01 | **0.97±0.01** |
| yacht | 308 | 6 | 1.82±0.01 | 1.83±0.01 | 1.82±0.01 | _1.77±0.01_ | **2.01±0.00** |
| year | 515345 | 90 | _3.54±NA_ | 3.55±NA | 3.55±NA | **3.60±NA** | 3.60±NA |
| **Average Rank** | | | 2.80±0.34 | 3.00±0.45 | **2.20±0.37** | 3.20±0.51 | 3.80±0.39 |

Table 2: Regression experiment: Average test RMSE

| Dataset | N | D | $\alpha \to -\infty$ | $\alpha = 0.0$ | $\alpha = 0.5$ | $\alpha = 1.0$ (**VI**) | $\alpha \to +\infty$ |
|---|---|---|---|---|---|---|---|
| boston | 506 | 13 | _2.84±0.18_ | 2.85±0.17 | 2.85±0.15 | **2.89±0.17** | 2.86±0.17 |
| concrete | 1030 | 8 | 5.28±0.10 | _5.24±0.11_ | 5.34±0.10 | **5.42±0.11** | 5.40±0.11 |
| energy | 768 | 8 | 0.79±0.04 | **0.88±0.05** | 0.81±0.06 | _0.51±0.01_ | 0.62±0.02 |
| naval | 11934 | 16 | 0.01±0.00 | 0.01±0.00 | **0.01±0.00** | _0.00±0.00_ | 0.00±0.00 |
| kin8nm | 8192 | 8 | **0.08±0.00** | 0.08±0.00 | 0.08±0.00 | 0.08±0.00 | _0.08±0.00_ |
| power | 9568 | 4 | 4.08±0.03 | **4.10±0.04** | _4.07±0.04_ | 4.07±0.04 | 4.08±0.04 |
| protein | 45730 | 9 | **4.57±0.05** | _4.44±0.03_ | 4.51±0.03 | 4.45±0.02 | 4.45±0.01 |
| wine | 1588 | 11 | 0.64±0.01 | **0.64±0.01** | 0.64±0.01 | 0.63±0.01 | _0.63±0.01_ |
| yacht | 308 | 6 | 1.12±0.09 | **1.24±0.11** | 1.11±0.08 | _0.81±0.05_ | 0.96±0.07 |
| year | 515345 | 90 | 8.95±NA | **9.13±NA** | 8.94±NA | 8.91±NA | _8.88±NA_ |
| **Average Rank** | | | 3.40±0.38 | 3.70±0.51 | 3.20±0.31 | 2.40±0.45 | **2.30±0.38** |

Figure 2: Sampled images from the the best models trained with IWAE (left) and VR-max (right).

Table 3: Network architecture of tested VAE algorithms.

| Dataset | L | architecture | activation | probability type (p/q) |
|---|---|---|---|---|
| Frey Face | 1 | d200-d200-s20 | softplus | Gaussian/Gaussian |
| Caltech 101 | 1 | d500-s200 | tanh | Bernoulli/Gaussian |
| MNIST & | 1 | d200-d200-s50 | tanh | Bernoulli/Gaussian |
| OMNIGLOT | 2 | d200-d200-s100-d100-d100-s50 | tanh | Bernoulli/Gaussian |