[Reviews · NeurIPS 2016]

Reviewer 1

Summary

The authors propose to use alpha-divergence in place of KL divergence in variational inference. They point out that alpha in [0,1] gives a lower bound, and alpha lt 0 an upper bound on the marginal likelihood. They suggest using MC to approximate the bound and give theoretical and experimental results on the bias of this approximation for alpha neq 1. They next derive two approaches to using minibatches, analogous to the previously derived stochastic EP and Black Box alpha-divergence approaches. Finally encouraging results are presented on neural network prediction on various UCI datasets and unsupervised learning with the variational autoencoder model.

Qualitative Assessment

The basic ideas here are sound, and it was nice to see some theory on how well MC can approximate the bound. In addition, the connections to SEP, BB-alpha and IWAE are intriguing. While these theoretical developments are nice, the paper could do more to convince one of the practical utility. The results in Fig 4 don't convince me to stop using straight VI. The theory suggests however that if I care about consistent parameter estimation or accurate marginal likelihood evaluation the proposed approaches might be very valuable. However, no empirical evidence of this is given. The results in Table 2 and Fig 6 show VR-max is close to IWAE in statistical performance, but it is only hinted at in the paper that VR-max will be computationally faster: this should be quantified explicitly. Some minor points on citations: I was surprised not to see "Divergence measures and message passing" Minka 2005 (Tech Report) being cited, since that's the first place I saw alpha-divergences (his notation is different). Also, both Variational Bayesian Inference with Stochastic Search (Paisley, Blei, Jordan) and Fixed-Form Variational Posterior Approximation through Stochastic Linear Regression (Salimans, Knowles) pre-date the black box VB paper in using MC to get gradients for VB. The later also introduced the "reparametrisation trick" beyond the Kingma/Welling VAE paper.

Confidence in this Review

3-Expert (read the paper in detail, know the area, quite certain of my opinion)


Reviewer 2

Summary

The paper proposes variational Renyi bound, a novel method for variational inference using Renyi divergence. The authors show how the proposed approach generalizes variational Bayes (VB) and importance weighted auto-encoders (IWAE). Finally, the authors illustrate the usefulness of the proposed approach using Bayesian neural networks and variational auto-encoders.

Qualitative Assessment

The paper is very nicely written and I enjoyed reading it. How was the log-likelihood computed for VR-max? If it's an upper bound (whereas VAE produces a lower bound), wouldn't it affect the comparisons? I think it'd be worth discussing the connections between Renyi divergence and alpha-divergence. In particular, the paper "Divergence measures and message passing" by Minka, 2005 discusses the connection between variational Bayes and expectation propagation in terms of alpha divergence. Minor issues: Figure 5: Hard to see the difference between K=5 and K=50 Figure 6b: figure too hard to read. page 7, line 222: RSME -> RMSE

Confidence in this Review

3-Expert (read the paper in detail, know the area, quite certain of my opinion)


Reviewer 3

Summary

Variational inference traditionally minimizes the Kullback-Leibler divergence from the approximating distribution to the target distribution. This paper studies the generalization of variational inference to Renyi's alpha-divergence, which yields the KL divergence for alpha=1. The paper studies the properties of the alpha-divergence with regards to upper and lower bounding the marginal likelihood, and describes and theoretically analyzes a stochastic strategy for minimizing it. Links with existing methods (such as variational autoencoders, importance weighted autoencoders, stochastic expectation propagation), which are special cases of the alpha-divergence framework, are discussed.

Qualitative Assessment

This is a very good and technically sound paper, containing a significant amount of material. The theoretical investigation of the properties of alpha-divergence minimization is thorough, clear and detailed. The paper provides significant theoretical insight and understanding into alpha-divergence minimization and optimization-based approximate inference in general. My biggest concern about the alpha-divergence framework is whether its theoretical richness and elegance actually translates to practical methods. In other words, I'm not sure that the practical aspects of it are appealing enough to convince practitioners of variational inference to switch to alpha-divergence minimization instead. I mainly view this paper as a very interesting and insightful investigation of alpha-divergence minimization, and less so as proposing a practical method. The results of the experiment on Bayesian learning of neural networks seem to suggest that in most cases the value of alpha makes little to no difference to the result. Also, there doesn't seem to be a consistent preference for a certain value of alpha; when there were any significant differences, they were mostly dataset dependent. This suggests that in practice alpha might end up being yet another parameter to tune. As a guide for choosing alpha, in lines 226-228 it is said that large alphas improve predictive error whereas small alphas improve log-likelihood. Even though this is what one would intuitively expect, I don't agree that such a trend can be seen in the experimental results. Equally off-putting might be the fact that for alphas far from 1 the stochastic gradients are heavily biased. If that in practice means drawing a large number of samples or using large minibatches and hence increasing computation time, then it might be preferable to stick with traditional KL minimization. I believe the most appealing practical aspect of alpha-divergence minimization is the opportunity for obtaining tighter lower (or possibly upper!) bounds to the marginal likelihood. As the paper discusses, this could improve maximum likelihood estimation of latent variable models with intractable likelihoods, such as the variational autoencoder. However, the heavy biasedness of the estimated likelihood bounds might in practice compromise this. An aspect which is not discussed in the paper is how varying alpha affects the difficulty of the resulting optimization problem. Varying alpha essentially changes the loss function, while its global optimum in the limit of infinitely flexible q remains the same. How does changing the loss function affect the optimization progress in practice? Is optimization for some alphas easier/harder, or is it roughly the same? Do some alphas lead to longer training times? If I understood correctly, in the Bayesian neural network experiment all networks were trained for the same number of iterations. Is this appropriate, or maybe for some alphas more training epochs should be used? Some minor comments follow: Table 1: for alpha->0, it should be supp(q) \subset supp(p), not the other way round. Figure 1(a): what do the contours correspond to? A fixed percentile? Do they all correspond to the same percentile? Line 130: I'd rather say "characterize the bias" or something similar. As far as I understand, theorem 2 doesn't provide a way to bound the bias. Line 147: "even though" instead of "even now"? Lines 186-187: I'd say something like "setting the gradient to zero will match the moments", rather than "the gradient will match the moments". Line 214: "neural network" instead of "neural networks". Line 222: "RMSE" instead of "RSME". Line 239: "do" instead of "does". Table 2: some OMNIGLOT log-likelihoods are meant to be negative instead? References section in gerenal: "Bayes", "Bayesian", "Boltzmann-Gibbs" and "Kullback-Leibler" should be capitalized.

Confidence in this Review

2-Confident (read it all; understood it all reasonably well)


Reviewer 4

Summary

A generalisation of Variational Inference is proposed to consider alpha-divergences, of which the KL-divergence is a special case. Making the parameter alpha vary produces different approximations with different properties. A bound (either upper or lower depending on the value of alpha) is derived, similarly to the classical variational lower-bound based on the KL-divergence. This bound can be approximated with a biased estimator (whose bias diminish with the number of samples used by the estimator) and a stochastic approximation of its gradient can be obtained. The effect of the value of alpha on posterior approximation is experimentally studied on Neural Network posterior approximation and variational auto-encoder training.

Qualitative Assessment

I found the paper interesting and straightforward to follow. The connection drawn with other algorithms are interesting insights. The question that I ask myself is of practical use: from Figure 4 or the additional results in the supplementary materials, it is hard to observe any significant difference in performance between different values of alpha or even if some value of alpha is consistently outperforming other. In that case, what is the motivation in essentially adding an additional hyper-parameter to a problem for no significant gain in performance? Some interrogations that I had while reading the paper: Section 3 - l. 103: Mode seeking (\alpha -> \infty). Does it really correspond to mode seeking? In the example of Figure 1. it might seem similar but wouldn't it correspond more to a zero forcing behaviour? (See Minka's "Divergence Measures and Message passing") What would the behaviour be when approximating a multi-modal distribution, such as a GMM? Section 4 - l.116: Why does the minimisation of the VR bound perform disastrously in MLE context? Section 4 - l.142: I don't understand how, for finite samples, negative alpha values can be used to improve the accuracy. It seems to me that as long as K is not infinity, there is no guarantee of E[L_{alpha, K}] to be an upper bound? Section 5 - l.258: You indicate that IWAE is inefficient as it wastes most of the computation on negligible samples, does it translate experimentally to runtime differences? Minor nitpicks/typos etc... - In table 1), The notes should be supp(q) \subseteq supp(p) (p and q reversed, unless I'm mistaken. This is the reverse as in theorem 1 because in Theorem 1, D_{\alpha}(q|p) is considered) - Line 65 imples->implies In supplementary: Proof of theorem 2: Some \prime are missing on the K.

Confidence in this Review

2-Confident (read it all; understood it all reasonably well)


Reviewer 5

Summary

Paper Summary ============= Conventional variational inference finds an approximate posterior that minimizes the KL divergence from the true (intractable) posterior. This paper shows how variational optimization objectives can use a wider class of Renyi's alpha divergences, which have a scalar parameter \alpha. Setting \alpha appropriately recovers many special cases of interest: \alpha -> 1 recovers the KL divergence while \alpha = 0 leads to exact posterior inference *if* the support of the approximating family contains the true posterior. Negative \alpha values provide an *upper* bound on marginal likelihood and tend to be "mass covering" as \alpha -> -infty, while postive \alpha values provide a lower-bound on the marginal likelihood and tend to be "mode seeking" as \alpha -> +infty. Aside from connecting Renyi's alpha-divergences to variational objectives (which I haven't seen before), this paper presents a useful algorithm for optimization of these objective functions based on Monte Carlo (MC) approximations similar to those used by Stan and black-box VI [11,12]. The algorithm works by drawing a handful of samples from the current *reparameterized* approximate posterior, computing importance weights for each sample, selecting one sample with probability according to its weight, and then stepping in the direction of its gradient. Special care must apply to minibatch learning, because samples are biased for values of \alpha other than \alpha = 1. Experiments show that alpha values often need to be tuned in a dataset-specific way for best performance, but that values other than alpha=1 are sometimes much better.

Qualitative Assessment

Review Summary ============== Overall, I recommend accepting this paper for at least a poster presentation. It nicely unifies and extends various variational inference approaches, and provides a good mix of theoretical and experimental contributions with some solid explanatory figures. My biggest concerns is a lack of practical description of how to properly choose the alpha value for a specific dataset, and how much this can matter compared to other hyperparameters that might need tuning (# of MC samples, sophistication of posterior approximation family etc). I worry that the question of "which alpha to use" will cause concern for practioners and will often result in setting alpha=1 in practice, which is the special case of KL-divergence we already know a lot about. I'd also like to see discussion of how different choices impact computational speed. Feedback on experiments ======================= My biggest concern is that choosing a good alpha value seems to need careful tuning for each dataset, and there isn't much guidance here on how to do it. It's not clear how much benefit we can get from properly tuning alpha, compared to say fixing alpha=1 (standard VI) and investing precious research time in other choices like using a larger number of samples for MC approximations, tuning minibatch sizes or learning rate schedules, or pursuing a slightly more sophisticated approximate posterior family. From a practical perspective, it would be nice to have a good default guess for alpha and some better intuition about how to tune it / when the default guess is "good enough". I found the overall story of the first experiment (with Bayesian neural networks) a bit confusing. I would have expected a relatively smooth quadratic-like trend in how the chosen alpha value affects performance, at least within a single dataset. That is, for any dataset, there is an obvious "best choice" of alpha, with performance getting worse for alpha values further away. owever, the Protein dataset has quite a weird story, where alpha=0.5 is noticeably worse than alpha = 0 or alpha = 1, and then alpha < 0 is worse again. Is there any explanation here? Might this be a local optima issue? In the supplement, the Energy and Yacht datasets have a similarly confusing story. I wish there was more practical discussion about how various algorithmic choices (# of samples K, alpha value, etc) impact the runtime of the algorithm.I'd be interested in both big-oh and (more importantly) wallclock time comparisons in the experiments. How important is it to just pick one MC sample to do the back-propagation, instead of using all of them? Feedback on Presentation ======================== I found Fig 2a (which gives intuition for the theorems quanitifying the bias in the MC approximations) quite hard to parse without lots of effort. It's not clear what the arrows mean. I'd suggest expanding this one figure into several smaller panels, each of which explores the influence of one parameter (# of samples K increasing, or alpha decreasing). Also, some other labeling like alpha_{+} or alpha_{-} might be more useful than alpha_1 and alpha_2 to visually emphasize the domains of these alpha values better. The technical discussion in Sec 4.3 explaining how to handle minibatches was quite confusing and hard to understand. I'd suggest a careful revision. Typos, etc ========== Fig 1 caption was hard to parse. Lines 192-202: Definitions of global and local as used here are a bit confusing. Could be clarified, since many readers will think of the global and local parameter classes described in SVI paper, which (I think) is a different sense of global/local.

Confidence in this Review

2-Confident (read it all; understood it all reasonably well)


Reviewer 6

Summary

The paper introduces variational inference with the KL divergence replaced by the more general Renyi divergence. The divergence has parameter alpha which can be used to control a smooth interpolation of the bound between existing lower bounds such as the ELBO and upper bounds on the marginal likelihood. This establishes a trade-off between mode-seeking and mass covering variational approximations. The bounds are optimized using monte carlo methods whose bias is analyzed in the paper. The authors introduce a new algorithm VR-max similar to IWAE where the algorithms always chooses the sample with the largest importance weight. Experiments investigate different choices of alpha for training a Bayesian neural network and compare VR-max to VAE and IWAE. It remains an open problem how to choose the best alpha for a given application.

Qualitative Assessment

Dear authors, thank you for the interesting read. I liked the idea of replacing the divergence measure with a more general form and thereby unifying various existing variational methods most notably VI and SEP. Neither its "derivation"/description nor the experimental results convinced me that VR-max is a useful algorithm. Why take the max? This no longer minimizes Renyi divergence. So what does it minimize? Also, in all but one of the experiments IWAE outperforms VR-max which is discouraging. My main criticism of the paper is its exposition. While it is full of good ideas, the big picture is often missing. For example, the contribution bullet points in the introduction promise an "optimization framework" while section 4 only mentions details such as how the reparametrization trick is used or how to make stochastic approximations without mentioning the big picture of the framework. Below are some more detailed comments and questions: l63: reformulate the proposition. It is not the definition which is continuous. l86: Here would be a good point to mention how the ELBO is optimized. Mention BBVI. l94: This is not a direct result l98 you mention the possibility of using a sandwich estimator. It would have been nice to see this point elaborated further in the experiments. l107 question about alpha=0: isn't "exact estimation" of log(p(D)) intractable? How was it obtained? l110 first paragraph of section 4. Clearly state how the bounds are going to be optimized. l129 question: why is the approximation biased? l169 why take the max? l197 question: what do you mean by local minimization in "EP minimisez alpha-divergence locally"? l206 if it is not clear how to choose alpha, why not just do VI (alpha=1)? l230 why does symmetry help? table2 how is the log likelihood for Frey Face positive? l260 is it the theory or the empirical results that justify VR-max?

Confidence in this Review

2-Confident (read it all; understood it all reasonably well)